# TRANSFORMERS ARE INHERENTLY SUCCINCT

**Pascal Bergsträßer**
RPTU Kaiserslautern-Landau
Kaiserslautern, Germany
bergstraesser@cs.uni-kl.de

**Ryan Cotterell**
ETH Zürich
Zurich, Switzerland
ryan.cotterell@inf.ethz.ch

**Anthony W. Lin**
RPTU Kaiserslautern-Landau and MPI-SWS
Kaiserslautern, Germany
lin@cs.uni-kl.de

## ABSTRACT

We study *succinctness* as a measure of the expressive power of transformers. Succinctness—how compactly a formalism can describe a language relative to other formalisms—is a classical notion in logic and automata theory. We prove that fixed-precision transformers are remarkably succinct: they can be exponentially more succinct than both linear temporal logic (LTL) and recurrent neural networks, and, by extension, state-space models, and doubly exponentially more succinct than finite automata. In other words, there exist families of languages describable by polynomial-size transformers whose smallest equivalent LTL formula or recurrent neural network is exponentially large, and whose smallest equivalent automaton is doubly exponentially large. We also establish matching upper bounds, showing that any fixed-precision transformer can be converted to an LTL formula with at most an exponential blow-up—improving a prior doubly exponential translation. As a consequence of this succinctness, we show that basic verification problems for transformers, such as emptiness and equivalence, are provably intractable: specifically, EXPSPACE-complete.

## 1 INTRODUCTION

Transformers (Vaswani et al., 2017) are the dominant architecture underlying most modern large language models. A substantial body of recent theoretical work has investigated their expressive power (Strobl et al., 2024; Barceló et al., 2024; Yang et al., 2024; Hahn, 2020; Pérez et al., 2021; Chiang and Cholak, 2022; Jerad et al., 2025), their trainability and ability to generalize to unseen strings of longer lengths (Zhou et al., 2024; Huang et al., 2025; Chiang and Cholak, 2022), and the extent to which their behavior can be formally verified (Sälzer et al., 2025). A key finding of this line of work is that transformers with finite precision—the setting most faithful to real-world hardware—recognize various classes of subregular languages depending on the exact assumptions made (Yang et al., 2024; Barceló et al., 2024; Jerad et al., 2025; Li and Cotterell, 2025).

Subregular languages constitute strict subclasses of the regular languages. For instance, the subregular class of star-free languages are precisely those definable by regular expressions that replace the Kleene star with intersection and complementation. The language $a^*b^*$ is star-free because it can be written as $\overline{\overline{\emptyset} \cdot b \cdot a \cdot \overline{\emptyset}}$, whereas $(aa)^*$ is not star-free (Straubing, 1994). By contrast, recurrent neural networks (RNNs) can recognize all regular languages under a fixed precision assumption (Minsky, 1967; Siegelmann and Sontag, 1995; Merrill et al., 2020; Svete and Cotterell, 2023), making them strictly more expressive than transformers as language recognizers. However, the strong empirical performance of transformers invites the question as to whether expressive capacity is the most revealing lens through which to compare architectures.

In this paper, we propose *succinctness* as an alternative lens for understanding the expressivity of transformers. The succinctness of a language $L$ with respect to a class $\mathcal{C}$ of language recognizers (e.g., transformers, finite automata, and formulas in $\mathsf{FO}[<]$) measures the size of the smallest

$C \in \mathcal{C}$ that recognizes $L$. In other words, succinctness tells us how many symbols are needed to describe $L$ with respect to the class $\mathcal{C}$. Succinctness is a classical notion in logic and computer science (Stockmeyer, 1974; Grohe and Schweikardt, 2004), where it sharpens expressive power into a complexity-theoretic refinement: rather than asking only which languages a formalism can recognize, succinctness asks how compactly each such language can be described within it. Greater succinctness comes at a price—more succinct formalisms typically have correspondingly harder decision problems, since their compact descriptions force any decision procedure to unfold a larger amount of underlying structure. A well-known example concerns linear temporal logic (LTL; Pnueli, 1977), which is expressively equivalent to the star-free languages (Libkin, 2004), and, hence, also to the counter-free automata of McNaughton and Papert (1971). Despite this equivalence in expressive power, LTL can be exponentially more succinct than finite automata (Sistla and Clarke, 1985), i.e., certain languages admit polynomial-size LTL formulas but require exponentially larger automata. A direct consequence is that decision problems for LTL, such as checking whether a formula recognizes a trivial language, are provably harder than the corresponding problems for automata (Sistla and Clarke, 1985).

This paper offers a formal result, which can be summarized as follows: transformers can describe certain languages extremely succinctly. Specifically, we show that transformers can be *exponentially* more succinct than LTL and RNNs, and hence also state-space models (SSMs; Gu and Dao, 2023; Merrill et al., 2024). Moreover, they are *doubly exponentially* more succinct than finite automata. In concrete terms, there exist families of languages describable by polynomial-size transformers that require exponentially larger LTL formulas or RNNs, and doubly exponentially larger automata. We also establish matching upper bounds: we give a translation from finite-precision transformers to LTL formulas of exponential size, significantly improving the doubly exponential translation of Yang et al. (2024). It follows that for any fixed-precision transformer, there is an equivalent LTL formula of exponential size and an equivalent finite automaton of doubly exponential size.[1] The key technical ingredient behind these results is showing that transformers can count from $0$ to $2^{2^N}$—that is, implement doubly exponentially large counters—via a subtle encoding using attention. We then prove that the resulting languages require exponentially larger descriptions as LTL formulas or RNNs, and doubly exponentially larger descriptions as finite automata. A natural consequence of this succinctness is that analyzing transformers should be computationally challenging. And, indeed, we show that checking whether a given transformer recognizes a trivial language, is EXPSPACE-complete. Under standard complexity-theoretic assumptions, this means that no algorithm can solve the problem in less than double exponential time.

The specific transformer model we study is the unique-hard attention transformer (UHAT), a simple and widely used abstraction of self-attention (Yang et al., 2024; Jerad et al., 2025; Strobl et al., 2024; Hao et al., 2022; Li and Cotterell, 2025; Hahn, 2020; Barceló et al., 2024; Bergsträßer et al., 2024). In particular, Jerad et al. (2025) show that expressivity bounds on UHATs entail corresponding bounds on softmax transformers with fixed precision. Different results in this paper hold under different precision assumptions: the UHAT upper bounds are stated for arbitrary rational weights, whereas the corresponding RNN results assume fixed (finite) precision. Importantly, this means our conclusions are valid in the setting that most faithfully mirrors real-world implementations—fixed precision arithmetic.

## 2 PRELIMINARIES

We adopt the following notational conventions in this paper. We write $\mathbb{N} \overset{\text{def}}{=} \{1, 2, 3, ...\}$ for the natural numbers and $\mathbb{N}_0 \overset{\text{def}}{=} \{0, 1, 2, 3, ...\}$ for the natural numbers including zero. Given $N \in \mathbb{N}$, we define $[N] \overset{\text{def}}{=} \{1, ..., N\}$. Furthermore, we write $\mathbb{Q}$ for the rational numbers. We denote scalars by lowercase italicized Latin letters, vectors by boldface lowercase italicized Latin letters, and matrices by boldface uppercase italicized Latin letters. For a vector $\boldsymbol{v} = (v_1, ..., v_D)$, we write $\boldsymbol{v}_{i:j} \overset{\text{def}}{=} (v_i, ..., v_j)$ for all $1 \leq i \leq j \leq D$, and $v_i$ for its $i^{\text{th}}$ component. An **alphabet** is a finite, non-empty set $\Sigma$ of **symbols**. A **word** (also called a **string**) is a finite sequence of symbols $\mathbf{a} = \mathrm{a}_1 \cdots \mathrm{a}_N$.

---

[1]This holds without exception: the LTL bound is the constructive translation of Prop. 13, and the automaton bound is its composition with the standard exponential LTL-to-automaton conversion (Sistla and Clarke, 1985; Vardi and Wolper, 1994); both are unconditional upper bounds on every fixed-precision transformer.

We denote symbols using lowercase Latin letters and words as boldfaced lowercase Latin letters. We write $|\mathbf{a}| = |\mathrm{a}_1 \cdots \mathrm{a}_N| = N$ for the length of a word $\mathbf{a}$. We write $\Sigma^*$ for the set of all **words**—including the empty word $\varepsilon$—and $\Sigma^+ \stackrel{\text{def}}{=} \Sigma^* \setminus \{\varepsilon\}$. A **language** is a subset $L \subseteq \Sigma^*$.

We assume familiarity with basic formal language theory and complexity theory; see Kozen (1997) and Sipser (1997) for standard references. In particular, we work with finite automata and the following complexity classes (Sipser, 1997):

$$\mathsf{P} \subseteq \mathsf{NP} \subseteq \mathsf{PSPACE} \subseteq \mathsf{EXP} \subseteq \mathsf{NEXP} \subseteq \mathsf{EXPSPACE}.$$

P and NP are problems solvable by a Turing machine in polynomial and nondeterministic polynomial time, respectively, and EXP and NEXP are their exponential-time counterparts. PSPACE and EXPSPACE are problems solvable by a Turing machine in polynomial and exponential space, respectively.

## 2.1 LINEAR TEMPORAL LOGIC

A formula in linear temporal logic (LTL) over an alphabet $\Sigma$ is defined by the grammar

$$\varphi ::= \top \mid \bot \mid Q_{\mathrm{a}} \, (\mathrm{a} \in \Sigma) \mid \varphi \wedge \varphi \mid \varphi \vee \varphi \mid \neg\varphi \mid \varphi \, \mathbf{S} \, \varphi \mid \varphi \, \mathbf{U} \, \varphi.$$

Satisfaction of an LTL formula $\varphi$ on a word $\mathbf{a} = \mathrm{a}_1 \cdots \mathrm{a}_N \in \Sigma^+$ at position $n \in [N]$, written $\mathbf{a}, n \models \varphi$, is defined inductively (omitting the trivial cases for $\top$ and $\bot$):

$$
\begin{aligned}
\mathbf{a}, n &\models Q_{\mathrm{a}} && \text{iff} && \mathrm{a}_n = \mathrm{a} && (\mathrm{a} \in \Sigma) \\
\mathbf{a}, n &\models \varphi_1 \wedge \varphi_2 && \text{iff} && \mathbf{a}, n \models \varphi_1 \text{ and } \mathbf{a}, n \models \varphi_2 \\
\mathbf{a}, n &\models \varphi_1 \vee \varphi_2 && \text{iff} && \mathbf{a}, n \models \varphi_1 \text{ or } \mathbf{a}, n \models \varphi_2 \\
\mathbf{a}, n &\models \neg\varphi_1 && \text{iff} && \mathbf{a}, n \not\models \varphi_1 \\
\mathbf{a}, n &\models \varphi_1 \, \mathbf{S} \, \varphi_2 && \text{iff} && \text{for some } j \text{ with } 1 \le j < n \colon \mathbf{a}, j \models \varphi_2 \text{ and} \\
& && && \text{for all } k \text{ with } j < k < n \colon \mathbf{a}, k \models \varphi_1 \\
\mathbf{a}, n &\models \varphi_1 \, \mathbf{U} \, \varphi_2 && \text{iff} && \text{for some } j \text{ with } n < j \le N \colon \mathbf{a}, j \models \varphi_2 \text{ and} \\
& && && \text{for all } k \text{ with } n < k < j \colon \mathbf{a}, k \models \varphi_1
\end{aligned}
$$

We also use the standard abbreviations

$$\mathbf{P}\varphi := \top \, \mathbf{S} \, \varphi \qquad \mathbf{F}\varphi := \top \, \mathbf{U} \, \varphi \qquad \mathbf{Y}\varphi := \bot \, \mathbf{S} \, \varphi \qquad \mathbf{H}\varphi := \varphi \wedge \neg\mathbf{P}\neg\varphi.$$

An LTL formula $\varphi$ recognizes the language $L(\varphi)$ consisting of all words $\mathbf{a} \in \Sigma^+$ where $\mathbf{a}, N \models \varphi$.[2]

**Example 1.** *The star-free language* $(\mathrm{ab})^+$ *can be defined in LTL as*

$$Q_{\mathrm{b}} \wedge \mathbf{H}\big(Q_{\mathrm{b}} \to \mathbf{Y}Q_{\mathrm{a}}\big) \wedge \mathbf{H}\big((Q_{\mathrm{a}} \wedge \mathbf{Y}\top) \to \mathbf{Y}Q_{\mathrm{b}}\big). \tag{1}$$

*Eq. (1) asserts that the last letter is* b*, every* b *is preceded by* a*, and every* a *that has a predecessor is preceded by* b*.*

## 2.2 UNIQUE-HARD ATTENTION TRANSFORMERS

**Symbol Embedding.** Let $\Sigma$ be an alphabet. A **symbol embedding** is a function $\mathrm{emb} \colon \Sigma \to \mathbb{Q}^D$ for some $D > 0$.[3] A symbol embedding naturally extends to a homomorphism $\Sigma^* \to (\mathbb{Q}^D)^*$, where $\mathrm{emb}(\mathrm{a}_1 \cdots \mathrm{a}_N) = \mathrm{emb}(\mathrm{a}_1), ..., \mathrm{emb}(\mathrm{a}_N)$ for $\mathrm{a}_1, ..., \mathrm{a}_N \in \Sigma$.

---

[2] For fragments that only allow $\mathbf{U}$ or $\mathbf{F}$, we use $\mathbf{a}, 1 \models \varphi$ instead.

[3] We define transformers over arbitrary rational numbers, as this is the most general setting in which our upper bounds hold. All results, however, carry over to fixed-precision arithmetic, i.e., a constant number of bits per value, independent of input length. The lower bounds hold under the even stronger restriction to fixed-precision integers. The precise statement of this carry-over depends on the formalization of fixed-precision arithmetic adopted: standard floating-point representations (Goldberg, 1991) are not associative, so the value of, for example, a dot product in an attention head can depend on the order in which its summands are evaluated. Our claim should therefore be understood with respect to a fixed, deterministic evaluation order; algebraic identities used in the proofs that rely on associativity (such as the order of summation) need to be re-checked under any other choice.

**Attention layer.** A unique hard-attention (UHA) layer of width $R > 0$ is specified by:

- Three affine transformations: $\boldsymbol{A}, \boldsymbol{B} \colon \mathbb{Q}^R \to \mathbb{Q}^R$ and $\boldsymbol{C} \colon (\mathbb{Q}^R \times \mathbb{Q}^R) \to \mathbb{Q}^S$;
- A **mask predicate** $M \colon \mathbb{N} \times \mathbb{N} \to \{0, 1\}$, defined as one of $M(n, m) \overset{\text{def}}{=} 1$ (no masking), $M(n, m) \overset{\text{def}}{=} \mathbf{1}[m < n]$ (strict future masking), or $M(n, m) \overset{\text{def}}{=} \mathbf{1}[m > n]$ (strict past masking);
- A **tie-breaking function** $\tau$ that selects an element of a finite, non-empty subset of $\mathbb{N}$, defined as either $\min$ (leftmost) or $\max$ (rightmost).

Given a sequence of $N$ vectors $\boldsymbol{v}_1, ..., \boldsymbol{v}_N \in \mathbb{Q}^R$ with $N \geq 1$, the layer operates as follows. The **score function** is defined as the dot product

$$\mathrm{S}(\boldsymbol{v}_n, \boldsymbol{v}_m) \overset{\text{def}}{=} \langle \boldsymbol{A}(\boldsymbol{v}_n), \boldsymbol{B}(\boldsymbol{v}_m) \rangle \tag{2}$$

for all $n, m \in [N]$. For each position $n \in [N]$, let

$$U_n \overset{\text{def}}{=} \{m \in [N] \mid M(n, m) = 1\} \tag{3a}$$

$$B_n \overset{\text{def}}{=} \{m \in U_n \mid \forall m' \in U_n \colon \mathrm{S}(\boldsymbol{v}_n, \boldsymbol{v}_m) \geq \mathrm{S}(\boldsymbol{v}_n, \boldsymbol{v}_{m'})\} \tag{3b}$$

be the set of unmasked positions and the subset of those that maximize the score, respectively. The **attention vector** at position $n$ is defined as $\boldsymbol{a}_n \overset{\text{def}}{=} \boldsymbol{v}_{\tau(B_n)}$ if $U_n \neq \emptyset$ and $\boldsymbol{a}_n \overset{\text{def}}{=} \boldsymbol{0}$ otherwise. The layer outputs the sequence $\boldsymbol{C}(\boldsymbol{v}_1, \boldsymbol{a}_1), ..., \boldsymbol{C}(\boldsymbol{v}_N, \boldsymbol{a}_N)$.

**ReLU layer.** A **ReLU layer** of width $R > 0$ applies, for a designated coordinate $r \in [R]$, the ReLU function to the $r^{\text{th}}$ component of each input vector. Formally, define $\rho_r \colon \mathbb{Q}^R \to \mathbb{Q}^R$ by

$$\rho_r(\boldsymbol{v}) \overset{\text{def}}{=} (\boldsymbol{v}_{1:r-1}, \max(0, v_r), \boldsymbol{v}_{r+1:R}). \tag{4}$$

Given a sequence of $N$ input vectors $\boldsymbol{v}_1, ..., \boldsymbol{v}_N \in \mathbb{Q}^R$ with $N \geq 1$, the layer outputs the sequence $\rho_r(\boldsymbol{v}_1), ..., \rho_r(\boldsymbol{v}_N)$ obtained by applying $\rho_r$ position-wise. Equivalently, one could place a feed-forward network at the end of each encoder layer (Hao et al., 2022; Barceló et al., 2024).

**Transformer.** A **unique hard-attention transformer (UHAT)** is a length-preserving function $\mathcal{T} \colon \Sigma^+ \to (\mathbb{Q}^S)^+$ obtained by composing a symbol embedding with a finite sequence of UHA and ReLU layers of conformable width. To use a UHAT $\mathcal{T} \colon \Sigma^+ \to (\mathbb{Q}^S)^+$ as a language recognizer, we equip it with an **acceptance vector** $\boldsymbol{t} \in \mathbb{Q}^S$. The language recognized by $\mathcal{T}$, denoted $L(\mathcal{T})$, consists of all words $\mathbf{a} \in \Sigma^+$ such that $\langle \boldsymbol{t}, \boldsymbol{v}_N \rangle > 0$ with $\mathcal{T}(\mathbf{a}) = \boldsymbol{v}_1, ..., \boldsymbol{v}_N \in \mathbb{Q}^S$.[4]

## 2.3 BOOLEAN RASP

As an intermediate step in proving EXPSPACE-hardness for UHATs, we use Boolean RASP (B-RASP; Yang et al., 2024), a programming language shown to be expressively equivalent to UHATs. A B-RASP program $\boldsymbol{P}$ is a finite sequence of predicates $P_1, ..., P_\Pi \in \{0, 1\}^{[N]}$. The program operates on an input word $\mathbf{a} = \mathrm{a}_1 \cdots \mathrm{a}_N \in \Sigma^+$. The first $|\Sigma|$ predicates are defined as follows. For each $\mathrm{a} \in \Sigma$, there is a lookup function $Q_\mathrm{a} \in \{0, 1\}^{[N]}$ defined by $Q_\mathrm{a}(n) = 1$ iff $\mathrm{a}_n = \mathrm{a}$. We label these predicates $P_1, ..., P_{|\Sigma|}$. Each remaining predicate $P_{t+1}$, for $t \geq |\Sigma|$, is built from $P_1, ..., P_t$ by one of two operations.

- A **position-wise operation** sets $P_{t+1}(i) := R(i)$, where $R(i)$ is a Boolean combination of $P_1(i), ..., P_t(i)$.
- An **attention operation** sets

$$P_{t+1}(i) := \blacklozenge_j \; [M(i, j), \mathrm{S}(i, j)] \, V(i, j) : D(i) \tag{5}$$

where $\blacklozenge \in \{\blacktriangleleft, \blacktriangleright\}$ and we define the following operations
- $\blacktriangleleft$ and $\blacktriangleright$ indicate leftmost and rightmost tie-breaking, respectively;
- $M(i, j)$ is a mask predicate as in the definition of a UHAT;
- $\mathrm{S}(i, j)$ and $V(i, j)$ are Boolean combinations of $P_1(i), ..., P_t(i)$ and $P_1(j), ..., P_t(j)$, called the **score predicate** and **value predicate**, respectively;

---

[4]For fragments that only allow strict past masking, we use $\langle \boldsymbol{t}, \boldsymbol{v}_1 \rangle$ instead.

- $D(i)$ is a Boolean combination of $P_1(i), ..., P_t(i)$.

The semantics of the attention operation are as follows. For each $i \in [N]$, let

$$o(i) \stackrel{\text{def}}{=} \begin{cases} \min\{j \in [N] \mid M(i, j) = 1 \text{ and } S(i, j) = 1\}, & \text{for } \blacktriangleleft \\ \max\{j \in [N] \mid M(i, j) = 1 \text{ and } S(i, j) = 1\}, & \text{for } \blacktriangleright . \end{cases} \tag{6}$$

Then $P_{t+1}(i) \stackrel{\text{def}}{=} V(i, o(i))$ if $o(i)$ exists, and $P_{t+1}(i) \stackrel{\text{def}}{=} D(i)$ otherwise.

We can view a B-RASP program as a language recognizer by asking whether $P_\Pi(N) = 1$.[5]

## 2.4 RECURRENT NEURAL NETWORKS

As with transformers, we treat recurrent neural networks as language acceptors, following Merrill et al. (2020) and Weiss et al. (2018; 2024). We define a **recurrent neural network** (**RNN**) as a quadruple $(\Sigma, g, \boldsymbol{h}_0, f)$ where $\Sigma$ is an alphabet, $g \colon (\mathbb{Q}^D \times \Sigma) \to \mathbb{Q}^D$ is a transition function, $\boldsymbol{h}_0 \in \mathbb{Q}^D$ is an initial hidden state, and $f \colon \mathbb{Q}^D \to \{\bot, \top\}$ is an acceptance function. Consider string $\mathbf{a} = \mathrm{a}_1 \cdots \mathrm{a}_N$. For $n \geq 1$, we define the $n^{\text{th}}$ hidden state $\boldsymbol{h}_n \stackrel{\text{def}}{=} g(\boldsymbol{h}_{n-1}, \mathrm{a}_n)$ inductively. We say $\mathbf{a}$ is accepted iff $f(\boldsymbol{h}_N) = \top$. As a computational model, it is natural to assume RNNs operate over a fixed precision, i.e., computation is always performed over rational numbers that can be represented with a constant $k$ number of bits. The details of the actual representation are not important for our analysis. Therefore, the state space of the above RNN can be mapped to $D$-vectors over $\{0, 1\}^k$ (instead of $\mathbb{Q}$). The following proposition is now immediate.

**Proposition 1.** *An RNN* $(\Sigma, g, \boldsymbol{h}_0, f)$ *with* $g \colon (\mathbb{Q}^D \times \Sigma) \to \mathbb{Q}^D$ *with fixed precision* $k$ *can be represented by a finite automaton with* $2^{kD}$ *many states.*

## 2.5 SIZE MEASURES AND SUCCINCTNESS

Let $\mathcal{R}$ be a finite representation of a language, i.e., in our case a UHAT, LTL formula, finite automaton, RNN, or B-RASP program. We define the size of $\mathcal{R}$, denoted by $|\mathcal{R}|$, as the length of its minimal binary encoding. In measuring succinctness of RNN, we put the precision $k$ in unary also as part of the size measure; since we do not want to compare a transformer that uses a fixed precision $k$ and allow an RNN that uses a fixed precision $2^k$.

**Definition 2** ($f$-more succinct). *Let* $\mathcal{C}^{(1)}$ *and* $\mathcal{C}^{(2)}$ *be classes of finite representations of languages, and let* $f \colon \mathbb{N} \to \mathbb{N}$ *be a function. We say* $\mathcal{C}^{(1)}$ *is* $f$-**more succinct** *than* $\mathcal{C}^{(2)}$ *if there is a family of languages* $\{L_n\}_{n=1}^\infty$ *together with representations* $\mathcal{R}_n^{(1)} \in \mathcal{C}^{(1)}$ *of* $L_n$ *such that every* $\mathcal{R}_n^{(2)} \in \mathcal{C}^{(2)}$ *representing* $L_n$ *satisfies* $|\mathcal{R}_n^{(2)}| \geq f(|\mathcal{R}_n^{(1)}|)$.

We say $\mathcal{C}^{(1)}$ is *exponentially* more succinct than $\mathcal{C}^{(2)}$ if it is $f$-more succinct for some $f(n) \in \Omega(2^{cn^d})$ with $c, d > 0$, and *doubly exponentially* more succinct if for some $f(n) \in \Omega(2^{2^{cn^d}})$ with $c, d > 0$.

**Definition 3** ($g$-bounded expansion). *Let* $\mathcal{C}^{(1)}$ *and* $\mathcal{C}^{(2)}$ *be classes of finite representations of languages, and let* $g \colon \mathbb{N} \to \mathbb{N}$ *be a function. We say* $\mathcal{C}^{(1)}$ *has* $g$-**bounded expansion** *over* $\mathcal{C}^{(2)}$ *if for every language* $L$ *and every choice of representation* $\mathcal{R}(2) \in \mathcal{C}^{(2)}$ *of* $L$, *there is a representation* $\mathcal{R}^{(1)} \in \mathcal{C}^{(1)}$ *of* $L$ *with* $|\mathcal{R}^{(1)}| \leq g(|\mathcal{R}^{(2)}|)$.

We say $\mathcal{C}^{(1)}$ has *polynomially* bounded expansion over $\mathcal{C}^{(2)}$ if it has $g$-bounded expansion for some polynomial $g$, and *exponentially* bounded expansion if for some $g(n) \in O(2^{cn^d})$ with $c, d > 0$.

Def. 2 and Def. 3 are duals. On one hand, Def. 2 is an *existential* lower-bound: it asks for a witness family on which $\mathcal{C}^{(2)}$ is forced to be at least $f$ times bigger than $\mathcal{C}^{(1)}$. On the other, Def. 3 is a *universal* upper-bound: it asks that on every language, every $\mathcal{C}^{(2)}$ representation has a $\mathcal{C}^{(1)}$ translation of size at most $g$. Neither definition alone is antisymmetric, but together they pin the gap: $\mathcal{C}^{(1)}$ is $f$-more succinct *and* has $g$-bounded expansion over $\mathcal{C}^{(2)}$ exactly when the size gap is at least $f$ on a witness family and at most $g$ uniformly.

---

[5]As for UHATs, we use $P_\Pi(1) = 1$ if only strict past masking is allowed.

## 3 THE SIZE OF SMALLEST WITNESS VIA NON-EMPTINESS PROBLEM

We now consider the problem of checking whether the language recognized by a UHAT or B-RASP program is non-empty. In particular, the technique is essentially a simulation of a Turing machine with an $2^{O(N)}$-sized tape for a given $N$. As we will see later, there are Turing machines such that the shortest accepted word by the constructed UHAT is of length at least $2^{2^{\Omega(N)}}$.

**Example 2.** *We consider an example with $N = 4$. Let $\Sigma = \{0, 1, \#, \mathrm{a}, \mathrm{b}, \mathrm{c}\}$, and let $H \stackrel{\text{def}}{=} \{(\mathrm{a},\mathrm{b}), (\mathrm{b},\mathrm{c}), (\mathrm{b},\mathrm{a}), (\mathrm{c},\mathrm{b})\}$ be a set of constraints specifying which symbols can appear in adjacent positions. We now describe a B-RASP program that accepts words of the form*

$$0000\mathrm{a}_1 \# 0001\mathrm{a}_2 \# 0010\mathrm{a}_3 \# \cdots \# 1111\mathrm{a}_{2^4} \#$$

*such that $(\mathrm{a}_n, \mathrm{a}_{n+1}) \in H$ for all $1 \le n < 2^4$. We show how to construct a B-RASP programs that (i) check that the bit counter is incremented, and (ii) check that the successive symbols are in $H$.[6] To check (i), we use the following attention operation:*

$$C_{+1}(i) := \blacktriangleright_j [j < i, Q_\#(j)] \bigvee_{k=1}^{4} \left( \bigwedge_{r=1}^{k-1} \left( \neg C_r(i) \wedge C_r(j) \right) \wedge C_k(i) \wedge \neg C_k(j) \right.$$
$$\left. \wedge \bigwedge_{r=k+1}^{4} \left( C_r(i) \leftrightarrow C_r(j) \right) \right) : 1 \tag{7}$$

*Assume $i$ is a $\#$-position. Attention selects the rightmost $\#$-position $j$ left of position $i$. Let $b_1^i \cdots b_4^i$ and $b_1^j \cdots b_4^j$ be the bit words directly left of position $i$ and $j$, respectively. We assume that we already defined $C_k(i) = b_k^i$ and $C_k(j) = b_k^j$ for all $k \in [4]$. Then, the above value predicate checks that the bit word $b_1^i \cdots b_4^i$ is the bit word $b_1^j \cdots b_4^j$ incremented by 1. To check (ii), we can use the attention operation*

$$M_\leftarrow(i) := \blacktriangleright_j [j < i, Q_\mathrm{a}(j) \vee Q_\mathrm{b}(j) \vee Q_\mathrm{c}(j)] \bigvee_{(\mathrm{h},\mathrm{h}') \in H} Q_\mathrm{h}(j) \wedge Q_{\mathrm{h}'}(i) : 1. \tag{8}$$

*If $i$ is a position of a symbol $\mathrm{a}_i$, attention picks the rightmost position $j$ of a symbol $\mathrm{a}_j$ to the left of $i$ and checks with the value predicate that $(\mathrm{a}_j, \mathrm{a}_i) \in H$. Two boundary conditions remain: the input must begin with the counter $0000$ and end with the counter $1111$. The first is a position-wise check at the leftmost $\#$, requiring $C_1(i) = \cdots = C_4(i) = 0$ at that position; the second is the analogous check at the rightmost $\#$, requiring all four bits to be $1$. We omit the construction of these gadgets here, since they follow the same pattern as $C_{+1}$ and $M_\leftarrow$.*

The construction given in Ex. 2 allows us to succinctly recognize a language whose shortest word has length exponential in the number of bits of the binary counter. In the following, we describe how to extend this idea such that we can reduce an EXPSPACE-complete problem to non-emptiness of a certain B-RASP program. Intuitively, we place multiple such words as above on top of each other, creating multiple rows and columns (separated by $\#$). Moreover, we introduce vertical constraints, i.e., between rows, in addition to the horizontal constraints $H$. Using this technique, we will see in Thm. 15 how B-RASP programs can even succinctly recognize languages whose shortest word has doubly exponential length.

Throughout the rest of this section, we build up to the proof of the following complexity bound.

**Theorem 4.** *The non-emptiness problem for UHATs and B-RASP programs is* EXPSPACE-*complete.*

To prove Thm. 4, we start with the lower bound for B-RASP programs.

**Proposition 5.** *The non-emptiness problem for B-RASP programs is* EXPSPACE-*hard.*

For the proof, we use the construction sketched in Ex. 2 and reduce from the tiling problem.

**Problem 6.** *We now describe the $2^N$-tiling problem. A tile is a quadruple $\boldsymbol{t} = \langle a, b, c, d \rangle \in \mathbb{N}_0^4$. We write $\mathsf{left}(\boldsymbol{t}) = a$, $\mathsf{up}(\boldsymbol{t}) = b$, $\mathsf{right}(\boldsymbol{t}) = c$, and $\mathsf{down}(\boldsymbol{t}) = d$.*

---

[6]Filling in the remainder of the B-RASP program to enforce the remaining constraints is straightforward.

**Given:** *An instance $\mathcal{I} = (N, T, \boldsymbol{t}_{fin})$, where $N > 0$ is an integer in unary, $T$ is a finite set of tiles, and $\boldsymbol{t}_{fin} \in T$ is a designated final tile.*

**Question:** *Do there exist a natural $M \in \mathbb{N}$ and a function $\tau \colon [2^N] \times [M] \to T$ such that*

1. *$\tau(2^N, M) = \boldsymbol{t}_{fin}$,*
2. *$\mathsf{down}(\tau(i, 1)) = \mathsf{up}(\tau(i, M)) = 0$ for all $1 \le i \le 2^N$,*
3. *$\mathsf{left}(\tau(1, j)) = \mathsf{right}(\tau(2^N, j)) = 0$ for all $1 \le j \le M$,*
4. *$\mathsf{right}(\tau(i, j)) = \mathsf{left}(\tau(i + 1, j))$ for all $1 \le i < 2^N$ and $1 \le j \le M$, and*
5. *$\mathsf{up}(\tau(i, j)) = \mathsf{down}(\tau(i, j + 1))$ for all $1 \le i \le 2^N$ and $1 \le j < M$?*

A configuration of tiles, i.e., a candidate for the function $\tau$, places tiles in $2^N$ columns and an arbitrary number ($M$) of rows.

**Proposition 7.** *The $2^N$-tiling problem is* EXPSPACE-*complete.*

*Proof.* The result follows from Theorem 5 in Schwarzentruber (2019) by choosing $k = 1$. ∎

To prove Prop. 5, we construct a B-RASP program of size polynomial in $N$ that accepts an encoding of a configuration of tiles as a sequence of words, similar to those displayed in Ex. 2, if and only if the configuration is a solution of the given $2^N$-tiling problem instance. The key observation is that strict future masking with rightmost tie-breaking enables us to check conditions between successive tiles in a row (Item 4) but also between the current tile and the tile at the most recent past occurrence of the same counter value, i.e., in the same column of the previous row (Item 5). The proof of the next lemma can be found in App. A.2.

**Lemma 8.** *Given a $2^N$-tiling problem instance, one can construct in time polynomial in $N$ a B-RASP program, whose language is non-empty iff the $2^N$-tiling problem instance has a solution.*

Lem. 8 reduces the $2^N$-tiling problem to the non-emptiness problem for B-RASP programs. Thus, together with Prop. 7, it implies Prop. 5.

We observe that the B-RASP program constructed in Lem. 8 is of a special form, which allows for a polynomial-time translation to UHAT.

**Lemma 9.** *Given a B-RASP program $P_1, ..., P_\Pi$ where every attention operation is of the form*

$$P_{t+1}(i) \coloneqq \blacklozenge_j \left[ M(i, j), \mathrm{S}(j) \wedge \bigwedge_{k \in K} P_k(i) \leftrightarrow P_k(j) \right] V(i, j) : D(i), \tag{9}$$

*where $|\Sigma| \le t < \Pi$, $\blacklozenge \in \{\blacktriangleleft, \blacktriangleright\}$, $\mathrm{S}(j)$ is a Boolean combinations of $P_1(j), ..., P_t(j)$, and $K \subseteq \{1, ..., t\}$, one can construct in polynomial time a UHAT that recognizes the same language.*

*Proof sketch.* Any Boolean combination of position-wise predicates can be simulated by a sequence of attention layers (each layer simply applies its affine map $\boldsymbol{C}$ to the current vector and disregards the attention vector) and ReLU layers. For each B-RASP attention operation, we use a single UHA layer (§ 2.2) whose mask predicate is $M$, whose tie-breaker matches $\blacklozenge$, whose affine maps $\boldsymbol{A}, \boldsymbol{B}$ compute the relevant components of the score, and whose affine map $\boldsymbol{C}$ combines the position's input $\boldsymbol{v}_n$ with the attention-selected $\boldsymbol{a}_n$. The value predicate $V(i, j)$ is simulated as follows: the layer's affine map $\boldsymbol{C}$ copies the relevant components of $\boldsymbol{a}_i$—the layer-$\ell$ vector that attention selected from position $o(i)$ (the unique position whose layer-$\ell$ vector wins the argmax in the UHA layer; equivalently, $o(i) = \tau(B_i)$ in the UHAT notation of § 2.2, with $B_i$ the argmax set among the unmasked positions and $\tau$ the tie-breaker)—into the layer's output at position $i$, and a small ReLU sub-network applies the Boolean combination $V$ to those copied components. The part $\mathrm{S}(j)$ of the score predicate that only depends on $j$ can be simulated using an additional preliminary layer that already computes the result of $\mathrm{S}(j)$ at every position $j$. For the part $\bigwedge_{k \in K} P_k(i) \leftrightarrow P_k(j)$ that checks equality of two binary numbers, we provide a score function that maximizes the attention score if the two binary numbers are equal. The full proof can be found in App. A.3. ∎

**Proposition 10.** *The non-emptiness problem for UHAT is* EXPSPACE-*hard.*

*Proof.* Together, Prop. 7, Lem. 8 and Lem. 9 imply the EXPSPACE lower bound for UHAT. ∎

**Corollary 11.** *The non-emptiness problem for UHATs in which every layer uses strict future masking and rightmost tie-breaking (or, dually, strict past masking and leftmost tie-breaking) is* EXPSPACE-*hard.*

*Proof.* The B-RASP program constructed in Lem. 8 uses only strict future masking and rightmost tie-breaking, and it can be adapted to use only strict past masking and leftmost tie-breaking.[7] The UHAT translation in Lem. 9 preserves the mask predicate and tie-breaking. Therefore the EXPSPACE lower bound established by Prop. 7, Lem. 8, and Lem. 9 transfers to UHATs in either of the two restricted classes. ∎

We now prove the upper bounds in Thm. 4. To this end, we first note that any B-RASP program can be converted in exponential time into an LTL formula using the construction given by Yang et al. (2024). In Prop. 13 we prove that the same holds true for UHATs, which improves the doubly exponential construction given by Yang et al. (2024) that translates UHATs into B-RASP programs first. These constructions suffice for the exponential-space upper bounds in Thm. 4 since non-emptiness of languages given by LTL formulas is in polynomial space (Sistla and Clarke, 1985).

To perform the translation from UHAT to LTL, we first have to make the crucial observation that the values occurring during the computation of a UHAT are not too large. The proof of the following proposition can be found in App. A.4.

**Proposition 12.** *For every UHAT $\mathcal{T}$, the precision required to evaluate $\mathcal{T}$ on any input is polynomial in $|\mathcal{T}|$, i.e., every rational value arising in the computation of $\mathcal{T}$ can be represented with at most* $\mathrm{poly}(|\mathcal{T}|)$ *bits.*

By Prop. 12, the set of rationals that can arise in the computation of $\mathcal{T}$ is finite, and each member has bit-length polynomial in $|\mathcal{T}|$. The set therefore has cardinality at most $2^{\mathrm{poly}(|\mathcal{T}|)}$, i.e., exponential in $|\mathcal{T}|$, and can be enumerated in exponential time. This is precisely what makes the layer-by-layer LTL construction in the proof of Prop. 13 feasible: at each layer we have a finite, enumerable, polynomial-bit-length set of vectors to range over, which implies that the LTL formula only has to simulate the position-wise behavior of attention layers, i.e., masking and selecting the position of the attention vector, but not the actual computation of values.

The proof of the following proposition can be found in App. A.5.

**Proposition 13.** *Given a UHAT $\mathcal{T}$ recognizing a language $L \subseteq \Sigma^+$, one can construct in exponential time an LTL formula $\varphi$ that recognizes $L$.*

Note, if we start with a UHAT, where every attention layer uses strict future masking and leftmost tie-breaking (resp. strict past masking and rightmost tie-breaking), then the LTL formula constructed in the proof of Prop. 13 only uses the **P** (resp. **F**) operator. It was shown by Sistla and Clarke (1985) that the non-emptiness problem for the fragments of LTL that only allow **P** or **F** is NP-complete. Thus, we obtain an improved complexity upper bound for such restricted UHATs.

**Corollary 14.** *The non-emptiness problem for UHATs, where each attention layer uses strict future masking/leftmost tie-breaking (resp. strict past masking/rightmost tie-breaking), is in* NEXP.

Note that it has been shown by Jerad et al. (2025) that such restricted UHATs are equally expressive as the LTL fragment with only **P** (respectively, **F**).[8] However, the construction by Jerad et al. (2025) from UHAT to the LTL fragments incurs a doubly exponential blow-up, as opposed to our singly exponential translation. The full proof of Thm. 4, combining the preceding lemmas, propositions, and corollaries into the two directions of EXPSPACE-completeness, can be found in App. A.1.

## 4 SUCCINCTNESS ACROSS REPRESENTATIONS

We now study how succinctly transformers can represent languages compared to standard models from formal language theory. We first compare transformers to LTL. One suggestion that trans-

---

[7]In case of strict past masking, we use the first coordinate in the acceptance condition.

[8]Note that LTL with a subset of the operators is often defined over EOS-padded strings; this choice affects its expressive capacity when using LTL with a subset of the operators. For instance, Li and Cotterell's (2026) demonstration that LTL[**P**] can *not* accept $\{a, b\}^* a$ with EOS-padding, but can without the padding.

formers may be more succinct than LTL comes from Thm. 4, which shows that the non-emptiness problem for UHATs is EXPSPACE-complete, whereas for LTL the corresponding problem is known to be PSPACE-complete. The following result shows that this exponential gap is also manifested in terms of the formalisms' respective succinctness.

**Theorem 15.** *UHATs are exponentially more succinct than LTL.*

*Proof.* It suffices to exhibit a witness family $\{L_n\}_{n=1}^{\infty}$ together with UHATs $\mathcal{T}_n$ of size $\mathrm{poly}(n)$ recognizing $L_n$ such that every LTL formula recognizing $L_n$ has size at least $c_1 2^{c_2 n}$ for constants $c_1, c_2 > 0$. Such a family is a witness in the sense of Def. 2: from $|\mathcal{T}_n| = \mathrm{poly}(n)$, say $|\mathcal{T}_n| \leq c_0 n^d$, we get $n \geq (|\mathcal{T}_n|/c_0)^{1/d}$, and substituting into $|\varphi_n| \geq c_1 2^{c_2 n}$ gives $|\varphi_n| \geq c_1 2^{c' |\mathcal{T}_n|^{1/d}}$ for $c_2' = c_2/c_0^{1/d}$. This witnesses $f$-more succinctness for $f(m) = c_1 2^{c_2' m^{1/d}}$ as required.

**Polynomial-size UHAT.** This direction proceeds in 3 steps.

1. Let $\mathcal{M}_n$ be a (deterministic) Turing machine that implements a binary counter with $2^n$ bits, i.e., it writes $0^{2^n}$ on its tape and increments the binary number until it has written $1^{2^n}$ on its tape and accepts. In particular, $\mathcal{M}_n$ first checks that it was started with the empty tape and then writes $2^n$ many 0's on its tape using an additional $n$-bit counter. To increment the $2^n$-bit counter, $\mathcal{M}_n$ traverses the counter from left to right while flipping every 1 to 0 until it encounters the first 0, which is then flipped to 1. To initialize the counter with 0's, $\mathcal{M}_n$ uses a linear number of states in $n$. Incrementing can be done with a constant-sized Turing machine. Moreover, $\mathcal{M}_n$ uses an exponential number of tape cells in $n$ and the unique accepting run has length at least $2^{2^n}$.
2. Van Emde Boas (1997) gives a reduction from Turing machines to tiling problem instances that encodes configurations of Turing machines in its rows and a correct tiling corresponds to a valid execution of the Turing machine. We observe that the $2^{p(n)}$-tiling problem instance $\mathcal{I}_n$, for some polynomial $p$, constructed from $\mathcal{M}_n$ has size polynomial in $n$ and it has the property that the smallest correct tiling has at least $2^{2^n}$ many rows.
3. Lem. 8 and Lem. 9 show that there is a UHAT $\mathcal{T}_n$ of size polynomial in the size of $\mathcal{I}_n$ that recognizes encodings of correct tilings of $\mathcal{I}_n$. Thus, $\mathcal{T}_n$ is of size polynomial in $n$ and the smallest accepted word has length at least $2^{2^n}$. We let $L_n$ be the language recognized by $\mathcal{T}_n$.

**Exponential LTL lower bound.** Let $\varphi_n$ be an LTL formula that recognizes $L_n$. Because the smallest accepted word by any LTL formula has length at most exponential in the formula size, using an exponential conversion from LTL to finite automata similar to Vardi and Wolper (1994), it follows that the size of $\varphi_n$ is at least exponential in $n$. ∎

Conversely, the next result shows that UHATs have polynomially bounded expansion over LTL: every LTL formula has an at most polynomially larger UHAT for the same language. Combined with Thm. 15, this pins the gap between UHATs and LTL: at most polynomial universally, and at least exponential on a witness family. The proof can be found in App. A.6.

**Proposition 16.** *UHATs have polynomially bounded expansion over LTL. In particular, given an LTL formula $\varphi$, one can construct in polynomial time a UHAT $\mathcal{T}$ such that $L(\mathcal{T}) = L(\varphi)$.*

We show next that UHATs are doubly exponentially more succinct than finite automata.

**Theorem 17.** *UHATs are doubly exponentially more succinct than finite automata.*

*Proof.* We reuse the witness family from Thm. 15: $L_n$ is the language recognized by the UHAT $\mathcal{T}_n$ constructed there. $\mathcal{T}_n$ is of size polynomial in $n$ and the smallest accepted word has length at least $2^{2^n}$. Because any automaton recognizing a non-empty language accepts a word of length at most linear in the automaton size, the smallest automaton that recognizes $L_n$ has size at least doubly exponential in $n$. Combined with $|\mathcal{T}_n| = \mathrm{poly}(n)$, this exhibits the witness family required by Def. 2 with $f$ doubly exponential. ∎

Conversely, the best known translation from counter-free automata—the class of finite automata equivalent to LTL—to LTL incurs an exponential blow-up (Maler and Pnueli, 1990). Composing this with Prop. 16 yields an at-most exponential-time translation from counter-free automata to

UHATs, which shows that UHATs have exponentially bounded expansion over finite automata when restricted to the star-free languages. The translation in Yang et al. (2024) also incurs an exponential blow-up via Maler and Pnueli (1990).

Combining Thm. 17 with Prop. 1 yields the following succinctness gap between UHATs and RNNs.

**Corollary 18.** *UHATs are exponentially more succinct than RNNs.*

## 5 APPLICATIONS

As a consequence of our results, we can show that reasoning about the language accepted by a UHAT, e.g., checking equivalence and emptiness, is intractable. Contrast this fact with deterministic finite automata, where these problems can be done in polynomial time (Kozen, 1997). As an example, we give a precise statement about the complexity of *equivalence problem*, i.e., the problem of checking whether two UHATs recognize the same language. The proof can be found in App. A.7.

**Theorem 19.** *Deciding the equivalence between two UHATs is* EXPSPACE-*complete.*

## 6 CONCLUDING REMARKS

**Related work.** Our work directly draws upon a number of recent results (Yang et al., 2024; Barceló et al., 2024; Jerad et al., 2025; Li and Cotterell, 2025), which demonstrate the close connection between unique-hard attention transformers and LTL and, thus, the star-free regular languages. However, none of these results concerns succinctness and computational complexity of verification. Closer to our complexity-theoretic angle, Sälzer et al. (2025) studied the verification problem for transformers of various precisions and showed that fixed-precision transformers are at least NEXP-hard (i.e., hard for the class of problems solvable by nondeterministic algorithms that run in exponential time). Their technique implies that transformers can be (singly) exponentially more succinct than finite automata, but yields no conclusion about their succinctness relative to representations like LTL or RNNs. Our results substantially improve on this by showing that transformers can be doubly exponentially more succinct than automata, and exponentially more succinct than LTL and RNNs. Our model is also simpler: we use unique-hard attention, whereas Sälzer et al. (2025) employs a combination of soft and hard attention. Our setting also restricts positional information to positional masking—a simple class of positional embeddings also considered by Yang et al. (2024), Jerad et al. (2025) and Li and Cotterell (2025)—in contrast to the arbitrary fixed-precision positional encodings admitted by Sälzer et al. (2025). Without position encodings, Sälzer et al. (2026) recently showed that verification is undecidable for average-hard-attention and softmax-attention transformers with finite but unbounded precision.

**Formal Verification of Transformers.** We close by situating our findings within the broader program of formal verification—the automated analysis, verification, and explanation of transformers—which is a central concern of explainable AI (Huang et al., 2020). Substantial practical progress has been made on verifying feed-forward neural networks, with tools developed over the last decade and benchmarked at the annual VNN competition (Brix et al., 2024); transformers, by contrast, remain largely out of reach. Despite the high worst-case complexity (EXPSPACE-complete), we pose as a challenge bringing techniques from automated verification (Clarke et al., 2018)—symbolic methods, simulation, *inter alia*—to bear on transformer verification in practice. Because our EXPSPACE-hardness proof requires transformers that encode large counters, a complementary direction is to identify subclasses that cannot encode such counters and thus admit lower-complexity verification. A related open question is the learnability of succinct transformers, on which the empirical evidence remains mixed (Garg et al., 2022; Naim et al., 2025; Huang et al., 2025). Finally, our results are a first step toward understanding how succinct transformers can be relative to other language-acceptor models, e.g., fixed-precision softmax transformers (Li and Cotterell, 2025), which UHATs overapproximate. We leave the succinctness of fixed-precision softmax and average-hard attention transformers as future work; see Yang et al. (2026) for an initial attempt.

ACKNOWLEDGMENTS

We thank David Chiang, Marco Sälzer, Andy Yang and anonymous reviewers for their helpful feedback. Pascal Bergsträßer and Anthony W. Lin are supported by Deutsche Forschungsgemeinschaft (grant number 522843867) and European Union[9] (ERC, LASD, 101089343).

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

# A  PROOFS

## A.1  PROOF OF THEOREM 4

**Theorem 4.** *The non-emptiness problem for UHATs and B-RASP programs is* EXPSPACE-*complete.*

*Proof.* The theorem asserts EXPSPACE-completeness of the non-emptiness problem for two formalisms simultaneously: B-RASP programs and UHATs. We prove the two directions of completeness separately.

**Hardness (lower bound).**  We establish EXPSPACE-hardness first for B-RASP programs, then transfer it to UHATs.

- **B-RASP**. For B-RASP programs, Prop. 7 states that the $2^N$-tiling problem is EXPSPACE-complete, so it is EXPSPACE-hard. By Lem. 8, any instance of the $2^N$-tiling problem can be transformed, in time polynomial in $N$, into a B-RASP program whose recognized language is non-empty if and only if the original tiling instance has a solution. Composing these two results yields a polynomial-time reduction from an EXPSPACE-hard problem to B-RASP non-emptiness. Thus, B-RASP non-emptiness is itself EXPSPACE-hard.
- **UHAT**. For UHATs, we further compose the previous reduction with the polynomial-time, language-preserving translation from B-RASP to UHATs of Lem. 9. The composed reduction is again polynomial-time, so the EXPSPACE-hardness of $2^N$-tiling transfers to UHAT non-emptiness; this is the content of Prop. 10.

**Membership (upper bound).**  We show that the non-emptiness problem for both formalisms lies in EXPSPACE by translating to LTL and invoking the Sistla–Clarke decision procedure.

For B-RASP programs, the construction of Yang et al. (2024) converts any B-RASP program $\boldsymbol{P}$ into an equivalent LTL formula $\varphi_{\boldsymbol{P}}$ in time exponential in $|\boldsymbol{P}|$; consequently $|\varphi_{\boldsymbol{P}}|$ is itself at most exponential in $|\boldsymbol{P}|$.

For UHATs, the analogous translation is supplied by Prop. 13, which turns any UHAT $\mathcal{T}$ into an equivalent LTL formula $\varphi_{\mathcal{T}}$ in exponential time (and hence of at most exponential size). The proof of Prop. 13 relies crucially on Prop. 12: the latter guarantees that all rational values arising in the computation of $\mathcal{T}$ admit representations of polynomially many bits, which is what makes the set of layer-wise representations enumerable in exponential time.

In both cases we have, in exponential time, reduced the original non-emptiness problem to LTL satisfiability for a formula of size at most exponential in the size of the original UHAT or B-RASP program. Sistla and Clarke (1985) prove that LTL satisfiability is decidable in space polynomial in the formula size; applied to $\varphi_{\boldsymbol{P}}$ or $\varphi_{\mathcal{T}}$, this uses space polynomial in an exponential quantity, that is, space exponential in $|\boldsymbol{P}|$ or $|\mathcal{T}|$. The overall decision procedure therefore runs in EXPSPACE.

Combining the two halves yields EXPSPACE-completeness of non-emptiness for both B-RASP programs and UHATs, as claimed. ∎

## A.2  PROOF OF LEMMA 8

**Lemma 8.** *Given a $2^N$-tiling problem instance, one can construct in time polynomial in $N$ a B-RASP program, whose language is non-empty iff the $2^N$-tiling problem instance has a solution.*

*Proof.* Fix a $2^N$-tiling problem instance $\mathcal{I} = (N, T, \boldsymbol{t}_{fin})$ as in Prob. 6. We construct a B-RASP program $\boldsymbol{P}_{\mathcal{I}}$ of size polynomial in $N$ and $|T|$ such that $L(\boldsymbol{P}_{\mathcal{I}}) \neq \emptyset$ if and only if $\mathcal{I}$ admits a solution $\tau\colon [2^N] \times [M] \to T$ in the sense of Prob. 6.

**Encoding.**  We encode a candidate $\tau$ as a word over the alphabet $\Sigma \coloneqq T \cup \{0, 1, \#\}$. Define the per-cell encoding $\mathrm{enc}\colon [2^N] \times [M] \to \Sigma^+$ by

$$\mathrm{enc}(i,j) \coloneqq \mathrm{bin}_N(i-1)\,\tau(i,j)\,\#, \tag{10}$$

where $\text{bin}_N(i-1) \in \{0,1\}^N$ is the $N$-bit binary representation of $i-1$, most significant bit first. The encoding of the full configuration scans columns within each row in order:

$$\text{enc}(\tau) := \text{enc}(1,1)\cdots\text{enc}(2^N,1)\,\text{enc}(1,2)\cdots\text{enc}(2^N,M). \tag{11}$$

Let $L_\mathcal{I} := \{\text{enc}(\tau) \mid \tau \text{ is a solution of } \mathcal{I}\}$. It suffices to construct a B-RASP program that recognizes $L_\mathcal{I}$.

**Plan.** Throughout, let $\mathbf{a} = a_1 \cdots a_L \in \Sigma^+$ denote the input, with $L := |\mathbf{a}|$. We build $\boldsymbol{P}_\mathcal{I}$ as a conjunction of five *gadgets*, each evaluated at the last position $L$:

- *Gadget A—shape*: $\mathbf{a} \in (\{0,1\}^N\,T\,\#)^*$.
- *Gadget B—column counter*: consecutive $N$-bit blocks count $0, 1, ..., 2^N - 1, 0, ...$.
- *Gadget C—final tile*: $\mathbf{a}$ ends with $1^N\,\boldsymbol{t}_{fin}\,\#$ (Item 1).
- *Gadget D—boundary conditions*: Item 2 and Item 3.
- *Gadget E—adjacency*: Item 4 and Item 5.

The output predicate is $Y(i) := A(i) \land B(i) \land C(i) \land D(i) \land E(i)$, and $\boldsymbol{P}_\mathcal{I}$ accepts $\mathbf{a}$ iff $Y(L) = 1$. We define each gadget below and verify its soundness at $L$.

**Gadget A: well-formed shape.** To check whether $\mathbf{a} \in (\{0,1\}^N\,T\,\#)^*$, we construct the following B-RASP predicates:

$$A_T(i) := \blacktriangleright_j [j < i, 1] \bigvee_{\boldsymbol{t} \in T} Q_{\boldsymbol{t}}(j) : 0 \tag{12a}$$

$$A_{\text{bit},1}(i) := \blacktriangleright_j [j < i, 1]\, Q_0(j) \lor Q_1(j) : 0 \tag{12b}$$

$$A_{\text{bit},k}(i) := \blacktriangleright_j [j < i, 1]\, A_{\text{bit},k-1}(j) : 0 \qquad \text{for } k = 2, ..., N \tag{12c}$$

$$A_{\#,1}(i) := \blacktriangleright_j [j < i, 1]\, Q_\#(j) : 1 \tag{12d}$$

$$A_{\#,k}(i) := \blacktriangleright_j [j < i, 1]\, A_{\#,k-1}(j) : 1 \qquad \text{for } k = 2, ..., N+1 \tag{12e}$$

$$A_{\text{enc}}(i) := (Q_\#(i) \to A_T(i)) \land \left( \left( \bigvee_{\boldsymbol{t} \in T} Q_{\boldsymbol{t}}(i) \right) \to \left( \bigwedge_{k=1}^N A_{\text{bit},k}(i) \right) \land A_{\#,N+1}(i) \right) \tag{12f}$$

We aggregate $A_{\text{enc}}$ across all positions:

$$A(i) := \blacktriangleright_j [j < i, \neg A_{\text{enc}}(j)]\, 0 : A_{\text{enc}}(i). \tag{13}$$

Then $A(L) = 1$ iff $A_{\text{enc}}(i) = 1$ at every position, which holds iff $\mathbf{a}$ matches $(\{0,1\}^N\,T\,\#)^*$ up to the requirement that the last symbol is $\#$, which is enforced by Gadget C below.

**Gadget B: column counter.** We check that the $N$-bit binary blocks separated by $\#$ encode the integers $0, 1, ..., 2^N - 1, 0, ...$ in order, generalizing the increment predicate of Ex. 2 from 4 bits to $N$ bits.

$$B_1(i) := \blacktriangleright_j [j < i, Q_0(j) \lor Q_1(j)]\, Q_1(j) : 0 \tag{14a}$$

$$B_k(i) := \blacktriangleright_j [j < i, Q_0(j) \lor Q_1(j)]\, B_{k-1}(j) : 0 \qquad \text{for } k = 2, ..., N \tag{14b}$$

$$B_{+1}(i) := \blacktriangleright_j [j < i, Q_\#(j)] \bigvee_{k=1}^N \left( \bigwedge_{r=1}^{k-1} \neg B_r(i) \land B_r(j) \right.$$

$$\left. \land\, B_k(i) \land \neg B_k(j) \land \bigwedge_{r=k+1}^N B_r(i) \leftrightarrow B_r(j) \right) : 0 \tag{14c}$$

$$B_{1\to 0}(i) := \blacktriangleright_j [j < i, Q_\#(j)] \bigwedge_{k=1}^N \neg B_k(i) \land B_k(j) : \bigwedge_{k=1}^N \neg B_k(i) \tag{14d}$$

$$B(i) := \blacktriangleright_j [j < i, Q_\#(j) \land \neg B_{1\to 0}(j) \land \neg B_{+1}(j)]\, 0 : B_{1\to 0}(i) \land B_{+1}(i) \tag{14e}$$

Then $B(L) = 1$ iff the binary blocks count up correctly.

**Gadget C: final tile.** For each tile $t \in T$, we record whether the most recent prior tile equals $t$ via the predicate $T_t$ at every position, and use this to check that $\mathbf{a}$ ends with $1^N t_{fin} \#$ (Item 1):

$$T_t(i) := \blacktriangleright_j \left[ j < i, \bigvee_{t' \in T} Q_{t'}(j) \right] Q_t(j) : 0 \qquad \text{for all } t \in T \tag{15a}$$

$$C(i) := Q_\#(i) \wedge T_{t_{fin}}(i) \wedge \bigwedge_{k=1}^{N} B_k(i) \tag{15b}$$

Then $C(L) = 1$ iff $\mathbf{a}$ ends with $1^N t_{fin} \#$.

**Gadget D: boundary conditions.** We enforce Item 2 and Item 3 of Prob. 6: the bottom row uses tiles with down $= 0$, the top row uses tiles with up $= 0$, and the leftmost (resp. rightmost) column uses tiles with left $= 0$ (resp. right $= 0$).

$$D_\perp(i) := \blacktriangleright_j \left[ j < i, Q_\#(j) \wedge \bigwedge_{k=1}^{N} B_k(i) \leftrightarrow B_k(j) \right] 1 : \bigvee_{\substack{t \in T, \\ \mathsf{down}(t)=0}} T_t(i) \tag{16a}$$

$$D_\top(i) := \blacktriangleright_j \left[ j < i, Q_\#(j) \wedge \left( \bigvee_{\substack{t \in T, \\ \mathsf{up}(t) \neq 0}} T_t(j) \vee \left( \bigwedge_{k=1}^{N} \neg B_k(j) \right) \right) \right]$$

$$\left( \bigvee_{\substack{t \in T, \\ \mathsf{up}(t)=0}} T_t(j) \right) \wedge \left( \bigvee_{\substack{t \in T, \\ \mathsf{up}(t)=0}} T_t(i) \right) : 0 \tag{16b}$$

$$D_\vdash(i) := \left( \bigwedge_{k=1}^{N} \neg B_k(i) \right) \rightarrow \bigvee_{\substack{t \in T, \\ \mathsf{left}(t)=0}} T_t(i) \tag{16c}$$

$$D_\dashv(i) := \left( \bigwedge_{k=1}^{N} B_k(i) \right) \rightarrow \bigvee_{\substack{t \in T, \\ \mathsf{right}(t)=0}} T_t(i) \tag{16d}$$

$$D(i) := \blacktriangleright_j \left[ j < i, Q_\#(j) \wedge \neg (D_\perp(j) \wedge D_\vdash(j) \wedge D_\dashv(j)) \right] 0 : D_\perp(i) \wedge D_\top(i) \wedge D_\vdash(i) \wedge D_\dashv(i) \tag{16e}$$

Then $D(L) = 1$ iff Item 2 and Item 3 hold.

**Gadget E: adjacency.** We enforce Item 4 and Item 5: horizontally adjacent tiles agree on (right, left), and vertically adjacent tiles (same column, consecutive rows) agree on (down, up) as follows:

$$E_\leftarrow(i) := \blacktriangleright_j \left[ j < i, Q_\#(j) \right] \left( \bigvee_{k=1}^{N} B_k(i) \right) \rightarrow \bigvee_{\substack{t, t' \in T, \\ \mathsf{left}(t)=\mathsf{right}(t')}} (T_t(i) \wedge T_{t'}(j)) : 1 \tag{17a}$$

$$E_\downarrow(i) := \blacktriangleright_j \left[ j < i, Q_\#(j) \wedge \bigwedge_{k=1}^{N} B_k(i) \leftrightarrow B_k(j) \right] \bigvee_{\substack{t, t' \in T, \\ \mathsf{down}(t)=\mathsf{up}(t')}} T_t(i) \wedge T_{t'}(j) : 1 \tag{17b}$$

$$E(i) := \blacktriangleright_j \left[ j < i, Q_\#(j) \wedge \neg (E_\downarrow(j) \wedge E_\leftarrow(j)) \right] 0 : E_\downarrow(i) \wedge E_\leftarrow(i) \tag{17c}$$

Then $E(L) = 1$ iff Item 4 and Item 5 hold.

**Wrap-up.** Define the output predicate

$$Y(i) := A(i) \wedge B(i) \wedge C(i) \wedge D(i) \wedge E(i), \tag{18}$$

and let $\boldsymbol{P}_{\mathcal{I}}$ accept iff $Y(L) = 1$. By the soundness of each gadget, $L(\boldsymbol{P}_{\mathcal{I}}) = L_{\mathcal{I}}$, so $L(\boldsymbol{P}_{\mathcal{I}}) \neq \emptyset$ iff $\mathcal{I}$ admits a solution. Each gadget uses $O(N)$ predicates of size polynomial in $N$ and $|T|$, so $|\boldsymbol{P}_{\mathcal{I}}|$ is polynomial in $N$ and $|T|$, hence in the size of $\mathcal{I}$. ∎

### A.3 PROOF OF LEMMA 9

**Lemma 9.** *Given a B-RASP program $P_1, ..., P_\Pi$ where every attention operation is of the form*

$$P_{t+1}(i) := \blacklozenge_j \left[ M(i,j), S(j) \wedge \bigwedge_{k \in K} P_k(i) \leftrightarrow P_k(j) \right] V(i,j) : D(i), \tag{9}$$

*where $|\Sigma| \leq t < \Pi$, $\blacklozenge \in \{\blacktriangleleft, \blacktriangleright\}$, $S(j)$ is a Boolean combinations of $P_1(j), ..., P_t(j)$, and $K \subseteq \{1, ..., t\}$, one can construct in polynomial time a UHAT that recognizes the same language.*

*Proof.* Let $\boldsymbol{P} = (P_1, ..., P_\Pi)$ be a B-RASP program over the alphabet $\Sigma = \{a_1, ..., a_{|\Sigma|}\}$, where $P_t$ is the initial vector $Q_{a_t}$ for all $1 \leq t \leq |\Sigma|$. We construct a UHAT over $\Sigma$ that recognizes the same language as $\boldsymbol{P}$. We use a one-hot symbol embedding $\mathsf{emb} \colon \Sigma \to \{0,1\}^{|\Sigma|}$, i.e., $\mathsf{emb}(a_t) := \boldsymbol{e}_t$ for all $1 \leq t \leq |\Sigma|$, where $\boldsymbol{e}_t$ denotes the $t^{\text{th}}$ unit vector. Then $P_t(i)$ coincides with the $t^{\text{th}}$ component of the $i^{\text{th}}$ input vector of the UHAT after the symbol embedding is applied. The UHAT will preserve these components in each layer and will gradually add new components to store the value of $P_t(i)$ for all $|\Sigma| < t \leq \Pi$. So assume we already defined the layers of the UHAT that compute the vector $(P_1(i), ..., P_t(i))$ at position $i$ for $|\Sigma| \leq t < \Pi$. We now define additional layers whose output will be $(P_1(i), ..., P_{t+1}(i))$.

We first consider the case where $P_{t+1}$ is a position-wise operation, i.e., $P_{t+1}(i)$ is defined by a Boolean combination of $P_1(i), ..., P_t(i)$. We define UHAT layers to compute the result of that Boolean combination bottom-up. Assume we already defined layers that output $(R_1(i), ..., R_s(i))$, where $s \geq t$ and $R_1(i), ..., R_s(i)$ contain $P_1(i), ..., P_t(i)$ and the results of previously computed subformulas. To compute the result of $\neg R_k(i)$ for some $k \in [s]$, we add an attention layer that just forwards $1 - R_k(i)$ at position $i$ in an additional component while leaving the first $s$ components unchanged. To compute $R_k(i) \wedge R_\ell(i)$ for some $k, \ell \in [s]$, we first use an attention layer to forward $R_k(i) + R_\ell(i) - 1$ in an additional component followed by a ReLU layer that forwards the result of $\max\{0, R_k(i) + R_\ell(i) - 1\}$ in this additional component, again leaving the first $s$ components unchanged. We do not have to deal with $R_k(i) \vee R_\ell(i)$, since it can be rewritten as $\neg(\neg R_k(i) \wedge \neg(R_\ell(i)))$. After computing the results of all subformulas, we add an additional attention layer to only forward $(P_1(i), ..., P_{t+1}(i))$, i.e., removing the intermediate results. Observe that a Boolean combination has only linearly many subformulas.

Let us now consider the case where $P_{t+1}$ is an attention operation of the form

$$P_{t+1}(i) := \blacklozenge_j \left[ M(i,j), S(j) \wedge \bigwedge_{k \in K} P_k(i) \leftrightarrow P_k(j) \right] V(i,j) : D(i) \tag{19}$$

as in the statement of Lem. 9. Throughout the construction below, $K$, $M$, $\blacklozenge$, $S$, $V$, and $D$ are the parameters of this input attention operation, as bound in the lemma statement; in particular, $K \subseteq \{1, ..., t\}$ is the index set of predicates whose values at $i$ and $j$ are compared for equality in the operation's score predicate. We first use additional layers as in the case of position-wise operations to compute the result of $\neg S(i)$ at every position $i$ in an additional component to output $(P_1(i), ..., P_t(i), 1 - S(i))$. Next we use an attention layer to add the result of $1 - P_k(i)$ for all $k \in K$ at every position $i$ in additional components. We then add an attention layer that uses mask predicate $M$, tie-breaking according to $\blacklozenge$, and attention score

$$\left( \sum_{k \in K} \left( P_k(i) P_k(j) + (1 - P_k(i))(1 - P_k(j)) \right) \right) - \left( 1 - S(j) \right) \tag{20}$$

which is equal to $|\{k \in K \mid P_k(i) = P_k(j)\}| - (1 - S(j))$ since

$$P_k(i) P_k(j) + (1 - P_k(i))(1 - P_k(j)) = \begin{cases} 1, & \text{if } P_k(i) = P_k(j) \\ 0, & \text{otherwise.} \end{cases} \tag{21}$$

Thus, the score is maximized (equal to $|K|$) if $P_k(i) = P_k(j)$ for all $k \in K$ and $S(j) = 1$. For every position $i$ let $o(i)$ be the position of the vector that maximizes the attention score with respect to $i$.

The attention layer forwards the vector $(P_1(i), ..., P_t(i), P_1(o(i)), ..., P_t(o(i)), \mathrm{S}(o(i)))$ at position $i$. We now compute the result of

$$R(i) := \mathrm{S}(o(i)) \wedge \bigwedge_{k \in K} P_k(i) \leftrightarrow P_k(o(i)) \tag{22}$$

at every position $i$ as in the case of position-wise operations and forward the vector $(P_1(i), ..., P_t(i), P_1(o(i)), ..., P_t(o(i)), R(i))$. Finally, we compute

$$\big(R(i) \wedge V(i, o(i))\big) \vee \big(\neg R(i) \wedge D(i)\big) \tag{23}$$

by again using additional layers as in the case of position-wise operations, whose result is exactly $P_{t+1}(i)$. We then forward $(P_1(i), ..., P_{t+1}(i))$.

It remains to describe when the UHAT accepts. If $P_t$ is the output vector of $\boldsymbol{P}$, then we stop the construction of the UHAT after the layers to compute $P_t$ are constructed. The acceptance vector of the UHAT is then defined as $\boldsymbol{e}_t$, i.e., the $t^{\text{th}}$ unit vector. This means that the UHAT accepts if and only if $\langle \boldsymbol{e}_t, (P_1(N), ..., P_t(N)) \rangle > 0$, which holds if and only if $P_t(N) = 1$, where $N$ is the length of the input.

We observe that the resulting UHAT only has polynomially many layers since the result of each operation $P_t$ can be computed using an additional number of layers that is linear in the description size of $P_t$. $\blacksquare$

### A.4 Proof of Proposition 12

**Proposition 12.** *For every UHAT $\mathcal{T}$, the precision required to evaluate $\mathcal{T}$ on any input is polynomial in $|\mathcal{T}|$, i.e., every rational value arising in the computation of $\mathcal{T}$ can be represented with at most $\mathrm{poly}(|\mathcal{T}|)$ bits.*

*Proof.* **Part 1: bounding the denominator.** Let $K$ be the number of rationals in the description of $\mathcal{T}$, i.e., the entries of the embedding $\mathrm{emb}(\mathrm{a})$ for $\mathrm{a} \in \Sigma$, the entries of every affine transformation $\boldsymbol{A}_\ell, \boldsymbol{B}_\ell, \boldsymbol{C}_\ell$ at each layer $\ell$, and the entries of the acceptance vector $\boldsymbol{t}$. Each of these rationals is part of $\mathcal{T}$'s binary encoding, so $K \leq |\mathcal{T}|$ and the bit-length $b$ of any individual rational satisfies $b \leq |\mathcal{T}|$ as well. Let $D$ be the least common multiple (LCM) of the denominators of those $K$ rationals. The LCM of $K$ integers each of bit-length at most $b$ has bit-length at most $K \cdot b$ (an upper bound is the product of the integers), so $D$ has bit-length at most $|\mathcal{T}|^2$. By construction, every embedding entry and every coefficient of an affine transformation can be written with denominator dividing $D$.

We now show by induction on the layer number $\ell$ that there is a denominator $d_\ell$, of polynomial bit-length, common to every value at layer $\ell$. The base case $\ell = 0$ is the embedding: $d_0 \stackrel{\text{def}}{=} D$ works. For the inductive step, suppose every layer-$\ell$ value has denominator dividing $d_\ell$. An attention layer takes an affine combination of layer-$\ell$ values, weighted by coefficients drawn from the description of $\mathcal{T}$. Each weight has denominator dividing $D$, and each input has denominator dividing $d_\ell$. The product of two fractions with denominators dividing $D$ and $d_\ell$ has denominator dividing $D \cdot d_\ell$; a sum of such products and the bias term (denominator dividing $D$, which divides $D \cdot d_\ell$) shares the same denominator. So we may take $d_{\ell+1} \stackrel{\text{def}}{=} D \cdot d_\ell$. A ReLU layer applies $\max\{0, \cdot\}$ pointwise, which leaves denominators unchanged, so $d_{\ell+1} \stackrel{\text{def}}{=} d_\ell$ suffices for that case. Iterating over at most $|\mathcal{T}|$ many layers, the final denominator is at most $D^{|\mathcal{T}|+1}$, with bit-length $(|\mathcal{T}| + 1) \log D$, polynomial in $|\mathcal{T}|$.

**Part 2: bounding the numerator.** At an attention layer of width $R$, each output numerator is an affine combination of $2R$ layer-$\ell$ numerators with integer weights and a bias each of bit-length at most $\log D$ (the entries of $\boldsymbol{C}$ scaled by $D$). The magnitude is therefore at most $2R \cdot 2^{\log D}$ times the largest layer-$\ell$ numerator's magnitude—an *additive* per-layer bit-length increase of $\log R + \log D + O(1)$; ReLU leaves bit-lengths unchanged. Since layer-0 numerators have bit-length at most $|\mathcal{T}| + \log D$, iterating across the at-most-$|\mathcal{T}|$ many layers gives a final bit-length of $O(|\mathcal{T}| \cdot (\log R + \log D))$, polynomial in $|\mathcal{T}|$.

Note that the additive (rather than multiplicative) per-layer growth above follows because the forwarded vector at an attention layer is $\boldsymbol{C}(\boldsymbol{v}_n, \boldsymbol{a}_n)$, an affine combination of two layer-$\ell$ vectors:

the position's own input $\boldsymbol{v}_n$ and the attention-selected $\boldsymbol{a}_n = \boldsymbol{v}_{\tau(B_n)}$, which is just a verbatim copy of whichever existing layer-$\ell$ vector $\tau$ picks. The score function $\mathrm{S}(\boldsymbol{v}_n, \cdot)$—including the dot product—does enter the picture and a single score value would have bit-length *quadratic* in its inputs. But the score is used only to rank positions; the argmax reduces it to a position index, and that index alone is used to fetch $\boldsymbol{a}_n$. The score's value never becomes a coordinate of any forwarded vector, so its quadratic bit-length never enters the next layer's input.

Combining the two parts: every value produced by $\mathcal{T}$ on any input has the form $p/q$ with $p$ and $q$ each of bit-length polynomial in $|\mathcal{T}|$. ∎

### A.5 PROOF OF PROPOSITION 13

**Proposition 13.** *Given a UHAT $\mathcal{T}$ recognizing a language $L \subseteq \Sigma^+$, one can construct in exponential time an LTL formula $\varphi$ that recognizes $L$.*

*Proof.* **Setup.** Let $\mathcal{T}$ be a UHAT recognizing a language $L \subseteq \Sigma^+$ and let $F$ be the set of binary representations of rational numbers that may occur during the computation of $\mathcal{T}$, as in Prop. 12. For the $\ell^{\text{th}}$ layer of $\mathcal{T}$ and every $\boldsymbol{v} \in F^S$, where $S$ is the output dimension of layer $\ell$, we construct an LTL formula $\varphi_{\boldsymbol{v}}^\ell$ such that, on input $\mathbf{a} = \mathrm{a}_1 \cdots \mathrm{a}_N \in \Sigma^+$, the $\ell^{\text{th}}$ layer outputs $\boldsymbol{v}$ at position $n \in [N]$ if and only if $\mathbf{a}, n \models \varphi_{\boldsymbol{v}}^\ell$. We define $\varphi_{\boldsymbol{v}}^\ell$ inductively on $\ell$.

**Base case ($\ell = 0$).** Let $\mathrm{emb}\colon \Sigma \to \mathbb{Q}^D$ be the symbol embedding of $\mathcal{T}$, and for all $\boldsymbol{v} \in F^D$ let

$$\varphi_{\boldsymbol{v}}^0 := \begin{cases} \displaystyle\bigvee_{\mathrm{a} \in \mathrm{emb}^{-1}(\boldsymbol{v})} Q_{\mathrm{a}}, & \text{if } \mathrm{emb}^{-1}(\boldsymbol{v}) \neq \emptyset \\ \bot, & \text{otherwise.} \end{cases} \tag{24}$$

We now define the formula for layer $\ell + 1$, splitting on the type of layer.

**Inductive step—ReLU layer.** If layer $\ell + 1$ is a ReLU layer of width $R$ applying ReLU to the $k^{\text{th}}$ coordinate, we set

$$\varphi_{\boldsymbol{v}}^{\ell+1} := \bigvee_{\substack{u \in F, \\ \max\{0, u\} = v_k}} \varphi_{(\boldsymbol{v}_{1:k-1}, u, \boldsymbol{v}_{k+1:R})}^\ell \tag{25}$$

for all $\boldsymbol{v} \in F^R$.

**Inductive step—attention with strict masking.** If layer $\ell + 1$ is an attention layer with strict future masking and rightmost tie-breaking defined by an affine transformation $\boldsymbol{C}\colon \mathbb{Q}^{2R} \to \mathbb{Q}^S$ and a score function $\mathrm{S}\colon \mathbb{Q}^{2R} \to \mathbb{Q}^R$, we let

$$\varphi_{\boldsymbol{v}}^{\ell+1} := \bigvee_{\substack{\boldsymbol{u}, \boldsymbol{a} \in F^R, \\ \boldsymbol{C}(\boldsymbol{u}, \boldsymbol{a}) = \boldsymbol{v}}} \varphi_{\boldsymbol{u}}^\ell \wedge \left( \left( \bigvee_{\substack{\boldsymbol{b} \in F^R, \\ \mathrm{S}(\boldsymbol{u}, \boldsymbol{b}) < \mathrm{S}(\boldsymbol{u}, \boldsymbol{a})}} \varphi_{\boldsymbol{b}}^\ell \right) \mathbf{S} \left( \varphi_{\boldsymbol{a}}^\ell \wedge \neg \mathbf{P} \bigvee_{\substack{\boldsymbol{b} \in F^R, \\ \mathrm{S}(\boldsymbol{u}, \boldsymbol{b}) > \mathrm{S}(\boldsymbol{u}, \boldsymbol{a})}} \varphi_{\boldsymbol{b}}^\ell \right) \right) \tag{26}$$

for all $\boldsymbol{v} \in F^S$. To account for the special case where the set of unmasked positions is empty, we take the disjunction of the previous formula with

$$(\neg \mathbf{P}\top) \wedge \bigvee_{\substack{\boldsymbol{u} \in F^R, \\ \boldsymbol{C}(\boldsymbol{u}, \boldsymbol{0}) = \boldsymbol{v}}} \varphi_{\boldsymbol{u}}^\ell. \tag{27}$$

We omit this special case in what follows. If the layer uses leftmost tie-breaking instead, we adapt the formula as follows:

$$\varphi_{\boldsymbol{v}}^{\ell+1} := \bigvee_{\substack{\boldsymbol{u}, \boldsymbol{a} \in F^R, \\ \boldsymbol{C}(\boldsymbol{u}, \boldsymbol{a}) = \boldsymbol{v}}} \varphi_{\boldsymbol{u}}^\ell \wedge \left( \mathbf{P} \left( \varphi_{\boldsymbol{a}}^\ell \wedge \neg \mathbf{P} \bigvee_{\substack{\boldsymbol{b} \in F^R, \\ \mathrm{S}(\boldsymbol{u}, \boldsymbol{b}) \geq \mathrm{S}(\boldsymbol{u}, \boldsymbol{a})}} \varphi_{\boldsymbol{b}}^\ell \right) \right) \wedge \left( \neg \mathbf{P} \bigvee_{\substack{\boldsymbol{b} \in F^R, \\ \mathrm{S}(\boldsymbol{u}, \boldsymbol{b}) > \mathrm{S}(\boldsymbol{u}, \boldsymbol{a})}} \varphi_{\boldsymbol{b}}^\ell \right) \tag{28}$$

The case of strict past masking is similar, with $\mathbf{U}$ in place of $\mathbf{S}$ and $\mathbf{F}$ in place of $\mathbf{P}$.

**Inductive step—attention without masking.** If the layer uses no masking and rightmost tie-breaking, we distinguish three cases according to where the attention vector lies relative to the current position: at the current position, strictly to the left, or strictly to the right. When the attention vector is at the current position, we define $\varphi_{\boldsymbol{v},\mathrm{at}}^{\ell+1}$ as

$$\varphi_{\boldsymbol{v},\mathrm{at}}^{\ell+1} := \bigvee_{\substack{\boldsymbol{u}\in F^R,\\ \boldsymbol{C}(\boldsymbol{u},\boldsymbol{u})=\boldsymbol{v}}} \varphi_{\boldsymbol{u}}^{\ell} \wedge \Big(\neg \mathbf{P} \bigvee_{\substack{\boldsymbol{b}\in F^R,\\ \mathrm{S}(\boldsymbol{u},\boldsymbol{b})>\mathrm{S}(\boldsymbol{u},\boldsymbol{u})}} \varphi_{\boldsymbol{b}}^{\ell}\Big) \wedge \Big(\neg \mathbf{F} \bigvee_{\substack{\boldsymbol{b}\in F^R,\\ \mathrm{S}(\boldsymbol{u},\boldsymbol{b})\geq \mathrm{S}(\boldsymbol{u},\boldsymbol{u})}} \varphi_{\boldsymbol{b}}^{\ell}\Big). \tag{29}$$

When the attention vector is strictly to the left of the current position, we define $\varphi_{\boldsymbol{v},\mathrm{L}}^{\ell+1}$ as

$$\varphi_{\boldsymbol{v},\mathrm{L}}^{\ell+1} := \bigvee_{\substack{\boldsymbol{u},\boldsymbol{a}\in F^R,\\ \boldsymbol{C}(\boldsymbol{u},\boldsymbol{a})=\boldsymbol{v},\\ \mathrm{S}(\boldsymbol{u},\boldsymbol{a})>\mathrm{S}(\boldsymbol{u},\boldsymbol{u})}} \varphi_{\boldsymbol{u}}^{\ell} \wedge \Big(\neg \mathbf{F} \bigvee_{\substack{\boldsymbol{b}\in F^R,\\ \mathrm{S}(\boldsymbol{u},\boldsymbol{b})\geq \mathrm{S}(\boldsymbol{u},\boldsymbol{a})}} \varphi_{\boldsymbol{b}}^{\ell}\Big) \wedge \Big(\Big(\bigvee_{\substack{\boldsymbol{b}\in F^R,\\ \mathrm{S}(\boldsymbol{u},\boldsymbol{b})<\mathrm{S}(\boldsymbol{u},\boldsymbol{a})}} \varphi_{\boldsymbol{b}}^{\ell}\Big) \mathbf{S} \Big(\varphi_{\boldsymbol{a}}^{\ell} \wedge \neg \mathbf{P} \bigvee_{\substack{\boldsymbol{b}\in F^R,\\ \mathrm{S}(\boldsymbol{u},\boldsymbol{b})>\mathrm{S}(\boldsymbol{u},\boldsymbol{a})}} \varphi_{\boldsymbol{b}}^{\ell}\Big)\Big). \tag{30}$$

Similarly, when the attention vector is strictly to the right, we define $\varphi_{\boldsymbol{v},\mathrm{R}}^{\ell+1}$ as

$$\varphi_{\boldsymbol{v},\mathrm{R}}^{\ell+1} := \bigvee_{\substack{\boldsymbol{u},\boldsymbol{a}\in F^R,\\ \boldsymbol{C}(\boldsymbol{u},\boldsymbol{a})=\boldsymbol{v},\\ \mathrm{S}(\boldsymbol{u},\boldsymbol{a})\geq \mathrm{S}(\boldsymbol{u},\boldsymbol{u})}} \varphi_{\boldsymbol{u}}^{\ell} \wedge \Big(\neg \mathbf{P} \bigvee_{\substack{\boldsymbol{b}\in F^R,\\ \mathrm{S}(\boldsymbol{u},\boldsymbol{b})>\mathrm{S}(\boldsymbol{u},\boldsymbol{a})}} \varphi_{\boldsymbol{b}}^{\ell}\Big) \wedge \Big(\mathbf{F}\Big(\varphi_{\boldsymbol{a}}^{\ell} \wedge \neg \mathbf{F} \bigvee_{\substack{\boldsymbol{b}\in F^R,\\ \mathrm{S}(\boldsymbol{u},\boldsymbol{b})\geq \mathrm{S}(\boldsymbol{u},\boldsymbol{a})}} \varphi_{\boldsymbol{b}}^{\ell}\Big)\Big) \wedge \Big(\neg \mathbf{F} \bigvee_{\substack{\boldsymbol{b}\in F^R,\\ \mathrm{S}(\boldsymbol{u},\boldsymbol{b})>\mathrm{S}(\boldsymbol{u},\boldsymbol{a})}} \varphi_{\boldsymbol{b}}^{\ell}\Big). \tag{31}$$

In the case of no masking and rightmost tie-breaking, we set

$$\varphi_{\boldsymbol{v}}^{\ell+1} := \varphi_{\boldsymbol{v},\mathrm{at}}^{\ell+1} \vee \varphi_{\boldsymbol{v},\mathrm{L}}^{\ell+1} \vee \varphi_{\boldsymbol{v},\mathrm{R}}^{\ell+1}. \tag{32}$$

The case of no masking and leftmost tie-breaking is analogous.

**Acceptance formula.** If $\mathcal{T}$ has $m$ layers whose last layer outputs vectors of dimension $S$, and $\boldsymbol{t} \in \mathbb{Q}^S$ is its acceptance vector, we define

$$\varphi := \bigvee_{\substack{\boldsymbol{v}\in F^S,\\ \langle \boldsymbol{t},\boldsymbol{v}\rangle>0}} \varphi_{\boldsymbol{v}}^m. \tag{33}$$

Then $\mathbf{a}, N \models \varphi$ if and only if $\mathbf{a} \in L$.

**Complexity.** It remains to argue that $\varphi$ can be computed in exponential time. By Prop. 12, $|F|$ is exponential in $|\mathcal{T}|$ and every representation in $F$ has polynomial bit-length; moreover, $F$ can be computed in exponential time. At every layer $\ell+1$ of width $R$, the formula $\varphi_{\boldsymbol{v}}^{\ell+1}$ depends on $|F|^{O(R)}$ formulas from layer $\ell$, and can be computed in time polynomial in $|F|^R \cdot |\mathcal{T}|$, since we only have to evaluate affine transformations on vectors from $F^R$, each of polynomial bit-length. At the last layer $m$, the formulas $\varphi_{\boldsymbol{v}}^m$ depend on $|F|^{O(R'm)}$ formulas from layer 0, where $R'$ is the maximum layer width. Hence $\varphi_{\boldsymbol{v}}^m$, and therefore $\varphi$, has size exponential in $|\mathcal{T}|$ and is computable in exponential time. ∎

### A.6 PROOF OF PROPOSITION 16

**Proposition 16.** *UHATs have polynomially bounded expansion over LTL. In particular, given an LTL formula $\varphi$, one can construct in polynomial time a UHAT $\mathcal{T}$ such that $L(\mathcal{T}) = L(\varphi)$.*

*Proof sketch.* We construct $\mathcal{T}$ by induction on the subformula structure of $\varphi$, maintaining the following invariant: after processing a subformula $\psi$, the UHAT built so far has, at every position $n$ of any input $\mathbf{a}$, a designated output coordinate carrying the truth value of $\mathbf{a}, n \models \psi$. Once the invariant is established for $\psi = \varphi$, the acceptance vector reads off that coordinate at the last position.

**Atomic formulas.** For $\varphi = Q_\mathrm{a}$, the token embedding already provides the indicator $\mathbf{1}[\mathrm{a}_n = \mathrm{a}]$ at each position, which is precisely the required truth value.

**Boolean combinations.** The truth value of $\neg\psi$, $\psi_1 \wedge \psi_2$, or $\psi_1 \vee \psi_2$ at position $n$ depends only on the children's truth values at the same position $n$, so a single affine-and-ReLU layer computes the required coordinate point-wise from coordinates produced by the inductive hypothesis.

**Since** ($\varphi = \varphi_1 \mathbf{S} \varphi_2$). By inductive hypothesis we have coordinates for $\varphi_1$ and $\varphi_2$ at every position; an affine-and-ReLU layer computes $\neg\varphi_1 \vee \varphi_2$ at every position. We then use an attention layer with strict future masking and rightmost tie-breaking to get for every position $i$ the maximal position $j < i$ where $\neg\varphi_1 \vee \varphi_2$ holds and output at position $i$ the truth value of $\varphi_2$ from position $j$. For positions $\ell$ and subformulas $\psi$, we write $\psi(\ell) \in \{0,1\}$ for the indicator of $\mathbf{a}, \ell \models \psi$. By the maximality of $j$, every $k$ with $j < k < i$ satisfies $\varphi_1(k) \wedge \neg\varphi_2(k)$ (otherwise $k$ would be a more recent witness); in particular $\varphi_1(k)$ holds for all $k \in (j, i)$. Thus if $\varphi_2(j) = 1$, then $j$ witnesses $\varphi_1 \mathbf{S} \varphi_2$ at $i$ under the semantics of § 2, and if $\varphi_2(j) = 0$ then $\neg\varphi_1(j) = 1$ blocks any earlier $j' \leq j$ from witnessing $\mathbf{S}$ at $i$. When no such $j$ exists, the attention layer's default value outputs 0, which is again consistent with the semantics.

The case where $\varphi = \varphi_1 \mathbf{U} \varphi_2$ is similar using strict past masking and leftmost tie-breaking. ∎

### A.7 PROOF OF THEOREM 19

**Theorem 19.** *Deciding the equivalence between two UHATs is* EXPSPACE-*complete.*

*Proof.* To prove the lower bound, we reduce from the non-emptiness problem for UHATs, which by Thm. 4 is EXPSPACE-complete. To this end, let $\mathcal{T}$ be a given UHAT and fix a UHAT $\mathcal{T}_0$ that recognizes the empty language. Then we have that $\mathcal{T}$ and $\mathcal{T}_0$ are equivalent if and only if $\mathcal{T}$ recognizes the empty language.

To prove the upper bound, let the UHATs $\mathcal{T}_1$ and $\mathcal{T}_2$ be given. We apply Prop. 13 to turn $\mathcal{T}_1$ and $\mathcal{T}_2$ in exponential time into LTL formulas $\varphi_1$ and $\varphi_2$, respectively. Now, $\mathcal{T}_1$ and $\mathcal{T}_2$ are equivalent if and only if $\varphi_1$ and $\varphi_2$ are equivalent. The latter can be decided in polynomial space (Sistla and Clarke, 1985), which results in an exponential-space algorithm in total. ∎

