# OpenReview forum: "Transformers are Inherently Succinct"
_ICLR.cc/2026/Conference — ICLR 2026 Oral_

### Official Review · Reviewer_TYFS · 2025-10-27

**Soundness:** 3
**Presentation:** 3
**Contribution:** 3
**Rating:** 8
**Confidence:** 4

**Summary:**

This paper gives a formal analysis of a certain class of transformers and shows that they can have very efficient codings for formal languages.  In particular, they can represent complex linguistic patterns that formal languages like LTL or neural models like RNNs require exponentially longer representations to express.

**Strengths:**

This is an important result that advances our understanding of the capacities of certain classes of transformers.   One upshot of this paper is that verifying equivalence of transformers, even relatively weak kinds, can be very difficult and computationally intractable.  This raises an interesting question for interpretability and explainability.   It its provably intractable to show that two transformers are equivalent, then we can't hope to get a computationally tractable method for determining whether a transformer is equivalent to some other model say expressed as an LTL formula.

**Weaknesses:**

When the authors give a lower bound for their result (theorem 5), they set it up following example 4.  Example 4 is interesting in this light: In that example we constructing a language L with s \in L and an attention layer that recognizes s.  Thus, example 4 provides a particular encoding and a particular attention layer set up.  What they show then is that transformers of the simple variety they define can accept a language for which the nonemptiness problem is EXSPACE hard.

Why is this important?  Well because of the implication the authors draw from this proof---namely that proving properties about transformers is computationally intractable.  Perhaps they simply mean "can be intractable", in which case there is no problem.  But transformers typically don't have such complex encodings (we know because we make those encodings when we do ICL and pretraining) and typically they are used to solve much easier problems.  Thus transformers can have high complexity but that doesn't mean that they do.  IFor instance, suppose you use a very simple transformer to learn something like the induction head phenomenon of Olsson et al 2022.  The attention layer structure may be quite simple and it can be straightforward to prove equivalence with other transformers or other models.  Thus, theorem 5 doesn't entail that proving properties of transformers is necessarily intractable.

It's also very unclear that transformers can actually learn the structure given in example 4 or the even more complex one that underpins theorem 5.  t will really depend on the task to which they are put.   What it does show, however, is that we don't have a tractable general method for proving properties for transformers.

A possible problem has to do here with the encoding of large numbers.  Transformers can have difficulty in doing even simple tasks given with large number or high vector norm inputs, inputs much larger than those seen in training, as work on in context learning has shown (Naim and Asher 2025).  If the authors could comment on this, it would be much appreciated.

Minor weaknesses:
It would help the reader to use the definition in Hao et al 2022 of these transformers as string accepters.

typos: show matching upper bound on ---> show a matching upper bound on

the first or last ---> the first or last

Please explain the black triangle notation when you introduce it in the specification of the Attention operation.

In example 4, Q I don't believe is defined.



Olsson, Catherine and Elhage, Nelson and Nanda, Neel and Joseph, Nicholas and DasSarma, Nova and Henighan, Tom and Mann, Ben and Askell, Amanda and Bai, Yuntao and Chen, Anna and others, arXiv:2209.11895, 2022

Naim, Omar and Asher,Nicholas. Analyzing limits of icl https://arxiv.org/pdf/2502.03503, 2025

**Questions:**

How does this result for UHATrelate to pushdown automata or stack automata for some of which the emptiness problem is PSPACE complete?  Can we place the UHAT class of transformers somewhere in this hierarchy?

---

> ### Author Response · Authors · 2025-11-19
> **Author Response**
>
> We thank the reviewer for taking the time to review our work. Revisions made in response to this review have been highlighted in orange in the updated version of the paper. We address the major ones below.
>
> **Comment/Question:**
> > Why is this important? Well because of the implication the authors draw from this proof ...
>
> **Answer:** Theorem 5 is a first result toward understanding the challenge in analyzing transformers. In particular, this double exponential time complexity is unavoidable in the **worst case**. Of course, this does not rule out the possibility of identifying a subclass of transformers, whose analysis is more tractable. We think that this is a highly interesting future research avenue, which we have added in the paper (see Concluding Remarks).
>
> **Comment/Question:**
> > It's also very unclear that transformers can actually learn the structure given in example 4 ...
> A possible problem has to do here with the encoding of large numbers ...
>
> **Answer:** This is a very intriguing question. In particular, as you correctly pinpointed, it is unclear if a training algorithm will always converge to such a succinct transformer (instead of a substantially larger one), even when the entire solution to the tiling problem is provided in the training data. In summary, we agree that the issue of learnability is at the moment unclear, and will need further investigation, building on the expressibility/succinctness results in our paper. We leave this for future work. In the new version, we have added a short discussion relating our work to learnability (see Concluding Remarks).
>
> **Comment/Question:**
> > How does this result for UHATrelate to pushdown automata or stack automata for some of which the emptiness problem is PSPACE complete? Can we place the UHAT class of transformers somewhere in this hierarchy?
>
> **Answer:** UHAT was shown by Yang et al. (2024) to be expressively equivalent to a strict subset of regular languages called star-free languages, which are in turn a strict subset of context-free languages (i.e. recognizable by pushdown automata). That said, recent works have studied stack attention (e.g. see https://arxiv.org/pdf/2310.01749), which allows transformers to recognize arbitrary context-free languages.

---

### Official Review · Reviewer_rC5n · 2025-11-01

**Soundness:** 3
**Presentation:** 3
**Contribution:** 3
**Rating:** 8
**Confidence:** 4

**Summary:**

The paper investigates the computational complexity of the emptiness problem for Unique Hard-Attention Transformer Models (UHAT). The authors show:

1. Non-Emptiness of UHATs is $\textsf{EXPSPACE}$-complete.
   They show the lower bound via an reduction from the $\textsf{EXPSPACE}$-complete $2^n$-Tiling problem.  The upper bound using a new translation from UHAT to LTL of single-exponential size, improving prior double-exponential constructions.
2. They show several succinctness results. Explicitly they show that UHATs can be exponentially more succinct than RNNs and LTL and doubly exponentially more succinct than finite automata.

As a consequence, several basic reasoning tasks (non-emptiness, equivalence, universality) for UHATs are $\textsf{EXPSPACE}$-complete.

**Strengths:**

- The results enrich our understanding of the attention mechanism in general and complement recent theoretical results on UHAT as well as on the emptiness problem for transformers with other attention mechanisms.
- The succinctness results and EXPSPACE bounds are technically non-trivial.
- Despite heavy theory, the paper is readable and provides good intuition

**Weaknesses:**

- UHAT is an intentionally minimal transformer abstraction. The paper would benefit from a discussion of how the results transfer to *practical architectures* (e.g., softmax attention, non-unique ties) and how much of the succinctness phenomenon is specific to UHAT
- The $\textsf{EXPSPACE}$-hardness construction (Prop. 6 & 8) relies on several non-trivial translations but omits the concrete UHAT layer-level construction. Without at least an appendix-level sketch (including an explicit bound on number of layers / parameters), the claimed polynomial reduction is difficult to verify.

This paper provides meaningful theoretical advances and improves important prior bounds. If the authors add a detailed construction of the EXPSPACE hardness reduction (Prop. 6/8), the contribution will be significantly stronger and easier to verify.

**Questions:**

1. It is interesting that for future-masking and left-most tiebreak (resp. past-masking and right-most tiebreak) the problem is in $\textsf{NEXP}$. The cause might be the lack of real positional information, right? Because after the maximum score was reached once, you will always attend to the first position which reached the score.
2. Lemma 9 shows equality checking of two binary strings via attention. In the proof of Proposition 6, value predicates check both equality and $Q_{\\#}(j)$. Is this covered by Lemma 9, or is an additional modification needed?
3. Can the authors include the explicit construction (or layer count) needed to simulate the B-RASP program of Prop. 8 as an UHAT? In particular, how do you avoid an exponential blow-up when translating Boolean expressions (e.g., no hidden DNF expansion as it was done in Yang et al. (2024))?

**Minor Comments**:

- In Section 2.4, the symbol $k$ is used both for embedding dimension and precision. Using distinct symbols would avoid confusion.

---

> ### Author Response · Authors · 2025-11-19
> **Author Response**
>
> We thank the reviewer for taking the time to review our work. Revisions made in response to this review have been highlighted in cyan in the updated version of the paper. We address the major ones below.
>
> **Comment/Question:**
> > UHAT is an intentionally minimal transformer abstraction. The paper would benefit from a discussion of how the results transfer to practical architectures ...
>
> **Answer:** This is an interesting question. Our results are one first step toward understanding how succinct transformers can represent languages compared to other language acceptor models. We initiate this investigation by focusing on UHATs. It would be very interesting to study similar questions in the case of other transformer models (like softmax or average hard attention). We have added this question to the future work section in the revised version of the paper.
>
> **Comment/Question:**
> > The EXPSPACE-hardness construction (Prop. 6 & 8) relies on several non-trivial translations but omits the concrete UHAT layer-level construction ...
>
> **Answer:** We have added a full proof of Proposition 8 to the appendix (Lemma 9 in the revised version). The full construction of the B-RASP program for Proposition 6 (Lemma 8 in the revised version) is also available in the appendix.
>
>
> **Comment/Question:**
> > It is interesting that for future-masking and left-most tiebreak (resp. past-masking and right-most tiebreak) the problem is in NEXP ...
>
> **Answer:** Exactly, this is the intuition why future masking with leftmost tie-braking is less expressive than with rightmost tie-breaking, since it corresponds to the LTL fragment that can only check whether something was true in the past (P operator), but no further positional information can be inferred from that. Equivalence to this LTL fragment was already observed by Jerad et al. (2025). With our singly exponential translation from these restricted UHATs to the LTL fragment and the fact that this fragment is NP-complete [Sistla & Clarke, 1985] we obtain the improved complexity bound NEXP.
>
> **Comment/Question:**
> > Lemma 9 shows equality checking of two binary strings via attention. In the proof of Proposition 6, value predicates check both equality and ...
>
> **Answer:** You are right. Technically, a slight modification of the mechanism in Lemma 9 is needed to incorporate the check $Q_{\\#} (j)$. One possibility is to subtract 1 from the attention score (which is maximized when the two binary strings are equal) if $Q_{\\#}(j)$ does not hold. However, since the crucial part is the equality check, we originally decided, for the sake of simplicity, to concentrate on this part in Lemma 9. In the revised version of the paper, Lemma 9 deals with this technicality and a full proof can be found in the appendix.
>
> **Comment/Question:**
> > Can the authors include the explicit construction (or layer count) needed to simulate the B-RASP program of Prop. 8 as an UHAT? ...
>
> **Answer:** We have added such an explicit constriction to the appendix (proof of Lemma 9 in the revised version). The main difference to the construction by Yang et al. (2024) is that we can assume B-RASP programs of a special form as constructed in Lemma 8 (of the revised version).

---

> > ### Comment · Reviewer_rC5n · 2025-11-27
> >
> > Thank you for the clarifications. All my questions and concerns have been sufficiently addressed. Therefore, I will keep my score.

---

### Official Review · Reviewer_cqAR · 2025-11-01

**Soundness:** 4
**Presentation:** 4
**Contribution:** 3
**Rating:** 8
**Confidence:** 3

**Summary:**

This paper studies the class of languages expressible by Uniform Hard Attention Transformers (UHAT) with fixed precision.

Previously, it has been shown that UHATs can recognize exactly the class of star-free languages, which is a subset of regular expressions. This is not as expressive as other classes, such as RNNs, which can recognize any regular language.

In order to understand potential benefits of UHATs over other classes of recognizers (such as Linear Temporal Logics (LTLs) and finite-state automata), this paper studies the succintness of this class. I.e., what languages can be recognized with polynomial-size UHATs.

The main results are:
(1) Determining whether a UHAT can recognize some string is EXPSPACE-complete.
(2) There is an exponential-time reduction from UHATs to LTLs.
(3) In fact, the above reduction is tight, in the sense that there are UHATs that require an exponentially-larger LTL to recognize the same language. And also any LTL can be reduced to a UHAT with a poly-time reduction.
(4) There are UHATs that require doubly-exponential size automata to recognize the same language. Therefore, they require exponentially larger RNNs to recognize the same language.

**Strengths:**

* This paper provides a rigorous and clear analysis of a special case of transformer models (UHATs). Upper bounds have complementary matching bounds. These help build a growing literature analyzing transformer architectures via a complexity-theoretic lens.

* The paper demonstrates that even though RNNs are more expressive than UHATs, there are languages that UHATs can recognize that would require an exponentially larger RNN to recognize. This is interesting, since it is a neat demonstration that expressivity is not the end of the story for comparing Transformer architectures to RNN architectures. Instead, the picture is more subtle once one studies what can be expressed efficiently (in poly-size).

* In my mind, this paper gives more motivation for hybrid architectures that incorporate both UHAT layers and RNN layers, getting the best of both worlds in terms of succinctness & expressivity.

**Weaknesses:**

* It is shown that UHATs are shown to have more succinct representations than RNNs for some languages. However, it is also known that RNNs are more expressive than UHATs. Therefore, in terms of succinctness of representations the two classes are incomparable. Thus, the title "Transformers are inherently succinct" is a little misleading, because RNNs are also "inherently succinct" at other languages that Transformers are not succinct at. A sentence acknowledging this (e.g. in the discussion or after Corollary 17) could go a long way.

**Questions:**

* See weaknesses.

---

> ### Author Response · Authors · 2025-11-19
> **Author Response**
>
> We thank the reviewer for taking the time to review our work. Revisions made in response to this review have been highlighted in green in the updated version of the paper.
>
> **Comment/Question:**
> > It is shown that UHATs are shown to have more succinct representations than RNNs for some languages. However, it is also known that RNNs are more expressive than UHATs ...
>
> **Answer:** Thank you for pointing this out. We agree with the observation and have added a clarification. In particular, UHATs can be exponentially more succinct than RNNs, among UHAT-expressible languages. We have added this remark, as well as that RNNs can recognize all regular languages (i.e. therefore are more expressive than UHATs), in the revised version of the paper below Corollary 18.

---

### Official Review · Reviewer_aJWw · 2025-11-10

**Soundness:** 2
**Presentation:** 2
**Contribution:** 3
**Rating:** 4
**Confidence:** 4

**Summary:**

This paper proposes *succinctness* as an alternative measure of the expressivity of the transformer architecture used in most LLMs. It goes on to prove that transformers (with finite precision, UHAT attention) can in fact be exponentially more succinct than LTL formulas and can be doubly-exponentially more succinct than traditional RNNs. They point out interesting corollaries, such as it being EXPSPACE-hard to solve the non-emptiness problem for UHAT transformers and for B-RASP programs; they do this by constructing languages whose shortest string is of (doubly-)exponential length in the size of the transformer / program.

While the idea of studying succinctness is creative and the results intriguing, diving deeper into the paper, I found many places where either the argument is not structured as clearly as it should have been, or where basic (but key) concepts are not described in clear detail, or where some leaps are made that need further justification.

Thus, while I am really excited to see this work be published, I feel it needs a careful pass by the authors to improve various aspects in order to make the paper more coherent and technically accurate. (see specific points below in the weaknesses section)

**Strengths:**

* The proposal of using "succintness" of representation as a way of measuring expressivity is novel in the study of the transformer architecture. It seem very interesting to pursue.

* The results (but see caveats below in the weaknesses section) are quite intriguing, showing how transformers can represent certain counting and tiling problems vastly more succinctly than LTL formulas and RNNs.

* The authors clearly identify the architectural assumptions (more like choices) early on in the paper, which helps place this work in the context of several alternatives that have been analyzed in the past few years. (but again see caveat in the weaknesses section below)

**Weaknesses:**

* While the paper makes it clear that the architectures use finite precision and unique rightmost hard attention, the authors don't mention other key aspects that seem important and consequential. E.g., the B-RASP program construction in the proof of Proposition 6 (tiling) uses $\Theta(n)$ predicates at each position $i$, which allows each # position to have access to the entire counter value represented by the previous $n$ positions, which then allows checking "locally" for increment by $1$. This is fine, but it also means the vectors represented in the transformer construction that simulates this program (in the proof of Theorem 14) need $\Theta(n)$ width vectors at each position. Besides making the studied transformer model non-uniform, it also means that when reconstructing the $\Theta(n)$ predicates at each # position, one needs $\Theta(n)$ attention "heads" to retrieve the past $n$ values into a single position, again adding to the non-uniform description of the transformer. Is this accurate? If so, it would be valuable to mention this early on -- that the transformer model (for the lower bound) will use linearly many attention heads and linear width. It should also be mentioned in section 2.2 where $d$ is introduced -- presumable $d$ isn't a constant, it's a linear function of $n$ in the constructions in this paper.

* In a similar vein, Proposition 11 seems to assume there is no *layer norm* or non-linearity like *sigmoid* or *inverse-tan* in the transformer architecture. Is this correct? If so, it would be good to note it somewhere.

* Some statements should be described more carefully. E.g., at the top of page 2 (line 54), it's not that transformers *are* exponentionally more succinct, it's that they *can be*, for certain problems (shown here for the $2^n$-tiling problem).

* Proposition 7 states that the $2^n$-tiling problem is EXPSPACE-complete, citing Schwarzentruber (2019). There is, however, no direct analog of this in the reference, from what I can tell. The closest I see is Theorem 5, which states that TILING$(2^n, n)$ is PSPACE-complete. The problem definition in that paper, however, has $n$ specified in **unary**. Are you deducing that, if $n$ were instead specified in **binary** (like you do), then TILING$(2^n, n)$ will become EXPSPACE-complete? This can use some clarification, also about the difference in the 2nd argument of the tiling problem. What you have appears closest to TILING$(2^n, *)$ in the notation of Schwarzentruber, and isn't a perfect match with a specific theorem of Schwarzentruber.

* The proof of Theorem 14 (arguably the main result) is confusing, to say the least:

* (a) The proof starts with a Turing machine $M_n$ that implements a binary counter with $2^n$ bits. It's stated that this machine has $poly(n)$ size without saying why. It seems that *given* a $2^n$ bit string, incrementing it needs only a fixed size Turing machine -- one just has to identify the rightmost $0$ and change it to $1$. The only complication is when the string looks like $0...01...1$, in which case the machine needs to detect that there are no $1$s to the left of the rightmost $0$, and change all $1$s to $0$s, which also seems doable without any machine size dependence on $n$. Is the $poly(n)$ size needed to write down $0^{2^n}$ at the very start?

* **[most critical issue]** (b) The proof says that the tiling problem instance $I_n$ constructed from $M_n$ has size $poly(n)$. Since the authors' definition of the $2^N$-tiling problem specifies the input $N$ in *binary* for tilings with $2^N$ rows (which means an instance with $2^N$ tiling rows has size $\log N$), this sounds correct with $N = 2^n$: input size is $O(n)$ and the tiling has $2^{2^n}$ rows, as the authors state. However, they then state that the proof of Prop 8 showed that there is a UHAT of size polynomial in the size of $I_n$, i.e., a UHAT of size $poly(n)$. First, the statement of Prop 8 is really about the non-emptiness problem being hard, not about the size of a UHAT; the latter is presumably buried inside the proof and should be brought out as a lemma that can be directly referenced in the proof of Theorem 14. Second, the proof of Prop 8 actually provides a polynomial time simulation of the B-RASP construction in the proof of Prop 6. So the size claim is actually coming from the proof of Prop 6 (again, the statement of Prop 6 can't be pointed to; one has to dig into its proof to find support for the $poly(n)$ size claim in the proof of Thm 14). This is where it seems to me that there is possibly a serious issue -- the B-RASP construction in the proof of Prop 6 constructs a program that has $\Theta(n)$ predicates at each position (as also noted above in my review), and, importantly, this is for an instance of the tiling problem of size $O(\log n)$. Specifically, when $n$ is provided as binary, the B-RASP program constructs at each position $i$ predicates $C_1, C_2, ..., C_n$. In other words, a tiling problem input with size $\log n$ requires B-RASP programs of size $\Theta(n)$ in Prop 6. So the tiling problem $I_n$, which has size $poly(n)$, will presumably require a B-RASP program of size $exp(n)$ by Prop 6, which implies UHATs of size $exp(n)$ by Prop 8, which conflicts with the argument presented in support of a $poly(n)$ size UHAT claim in Theorem 14 in lines 398-401.

This last part -- the proof of Theorem 14 -- is in the best case confusing and in the worst case incorrect. Again, I like where this paper is going and it's possible that the results are actually correct. But the current way the arguments are presented is confusing and, to be honest, somewhat sloppy in places. Having clear statements of "nuggets" of proofs that are cited and used later would have been much better, rather than indirections through multiple proofs.


* Clarification: in section 2.5 (size measures), why is succinctness not defined directly as $|R_2(n)| = 2^{\Omega(|R_1(n)|)}$? Does one have to go through every $f \in 2^{o(n)}$? (this might be true, would help to explain if so)

* The paragraph at the bottom of Example 4 ("This allows us to succinctly ...") is unclear when it talks about "stacking multiple such strings vertically... double exponential length". Please clarify what this means and how this will be used later in the paper.

**Questions:**

Please see the lengthy discussion above, especially around the proof of Theorem 14 (which builds upon subtleties in the proofs of Prop 6 and the definition the tiling problem considered here). I am willing to raise my score if the authors are able to address these concerns clearly during the rebuttal phase.

---

> ### Author Response · Authors · 2025-11-19
> **Author Response (1/2)**
>
> We thank the reviewer for the very detailed review. Revisions made in response to this review have been highlighted in yellow in the updated version of the paper. We address the major ones below.
>
> **Comment/Question:**
> > While the paper makes it clear that the architectures use finite precision and unique rightmost hard attention, the authors don't mention other key aspects that seem important and consequential ...
>
> **Answer:** Here the difference between an instance of the tiling problem and the actual input of the UHAT/B-RASP is important. The former contains the number $n$ in unary and a set of tiles. The latter is an encoding of the function $\tau$ in the tiling definition, i.e., a description of a specific configuration (with rows and columns) of tiles. In Proposition 6 we provide a construction from a tiling problem instance to a B-RASP program, whose size is polynomial in $n$ (which is given in unary). It is correct that this program uses attention to gather all the bits of one counter. Such counters are used to number the columns in each row, i.e., they use $n$ bits in the $2^n$-tiling problem that only allows configurations with $2^n$ columns. So the B-RASP program needs linearly many operations in $n$ to gather these bits. But it is important to observe that this number $n$ does not grow with the input (configuration) length, which can be unbounded since the number of rows is unbounded. Thus, $n$ is fixed for a fixed tiling problem instance. This means, given a $2^n$-tiling problem instance, we construct one B-RASP program that is of size polynomial in $n$ that accepts all encodings of valid tiling configurations.
> Similarly, the UHAT constructed in Theorem 14 only needs a linear number of layers in $n$ with in our definition only a single attention head per layer, as opposed to a variable number of attention heads that depends on the input length. Thus, if “non-uniform” means that parameters of the UHAT layers depend on the input length, then our UHAT model is uniform, since $n$ is fixed. Moreover, the number $d$ is fixed for a fixed tiling problem instance.
> We have added an explanation of the difference between tiling problem instance and tiling configuration to the revised paper below the definition of a $2^n$-tiling problem.
>
> **Comment/Question:**
> > In a similar vein, Proposition 11 seems to assume there is no layer norm or non-linearity like sigmoid or inverse-tan in the transformer architecture. Is this correct? If so, it would be good to note it somewhere.
>
> **Answer:** That is correct: UHAT uses affine transformations and piecewise linear activation functions (like ReLU), but no layer norm and non-linear activation functions like sigmoid and inverse-tan. We use the UHAT transformer model for two reasons. Firstly, it is a simple (i.e. amenable to theoretical analysis) and popular abstraction of self-attention (e.g. see Hao et al. (2022), Yang et al. (2024)). Secondly, from the perspective of expressivity, it is known (e.g. see Jerad et al. 2025) that bounds on UHAT entail bounds on realistic softmax transformers operating at fixed precision. Whether similar conclusions can be drawn in the case of succinctness is a very interesting question for future research. We clarify this in the new version (see paragraph on assumption in introduction and concluding remarks).
>
> **Comment/Question:**
> > Some statements should be described more carefully. E.g., at the top of page 2 (line 54), it's not that transformers are exponentionally more succinct, it's that they can be, for certain problems (shown here for the $2^n$-tiling problem).
>
> **Answer:** Thank you for pointing this out. In the revised version we wrote “can be more succinct” as we already did in the theorems.
>
> **Comment/Question:**
> > Proposition 7 states that the $2^n$-tiling problem is EXPSPACE-complete, citing Schwarzentruber (2019). There is, however, no direct analog of this in the reference, from what I can tell ...
>
> **Answer:** Proposition 7 appeared as Theorem 5 in [Schwarzentruber, 2019]. There it says that TILING$(\*,\text{exp}_k(n))$ is $k$-EXPSPACE-complete. In our case we choose $k=1$. Note that TILING$(\*,\text{exp}(n))$ in [Schwarzentruber, 2019] refers to the problem where the number of rows is unbounded (indicated by *) and the number of columns is $2^n$. Thus, the problem coincides with what we call $2^n$-tiling (the only difference is that for convenience we assume a final tile is given instead of an initial tile, which is interreducible). Here, it is important that we also assume $n$ to be given in unary (as mentioned in the “Given” part of our tiling problem definition). In the revised version of the paper we pinpoint where in [Schwarzentruber, 2019] the statement of Proposition 7 can be found.

---

> > ### Author Response · Authors · 2025-11-19
> > **Author Response (2/2)**
> >
> > **Comment/Question:**
> > > The proof of Theorem 14 (arguably the main result) is confusing, to say the least:
> > (a) The proof starts with a Turing machine $M_n$ that implements a binary counter with $2^n$ bits ...
> >
> > **Answer:**  Indeed, incrementing the counter can be done with a constant-sized Turing machine. Since we want the machine to implement a counter with $2^n$ bits, which ensures that it uses an exponential number of tape cells in $n$ and the accepting execution has length doubly exponential in $n$, we first have to check that the Turing machine is initialized with $0^{2^n}$ or, as mentioned in the review, write down $0^{2^n}$ first if we start with the empty tape. To implement this initial check/task, the Turing machine can maintain an additional counter on its tape with $n$ bits. Making sure that this additional counter uses exactly $n$ bits requires $n$ states in the Turing machine. Thus, this Turing machine has poly$(n)$ size, which is just an upper bound that suffices for our purposes. We have added a clarification to the proof of Theorem 15 in the revised version of the paper.
> >
> > **Comment/Question:**
> > > **[most critical issue]** (b) The proof says that the tiling problem instance $I_n$ constructed from $M_n$ has size poly$(n)$ ...
> >
> > **Answer:** As mentioned above, we assume $n$ in the definition of the $2^n$-tiling problem to be given in **unary** (not binary as stated in the review). The tiling problem instance $I_n$ in Theorem 14 that is constructed from the Turing machine $M_n$ is of size polynomial in $n$, i.e., $I_n$ is a $2^{\text{poly}(n)}$-tiling problem instance where the number poly$(n)$ encoded in unary is part of the instance $I_n$. In the paper we say that $I_n$ is a $2^n$-tiling instance, but this should be $2^{\text{poly}(n)}$ since the Turing machine $M_n$ does not use exactly $2^n$ many tape cells (we corrected this in the revised version). The proof of Proposition 6 describes how to construct a B-RASP program from a tiling problem instance, like $I_n$, that is polynomial in the size of the instance, in this case polynomial in the size of $I_n$, which is polynomial in $n$. Thus, the B-RASP program constructed from $I_n$ has size polynomial in $n$. The proof of Proposition 8 then describes how this B-RASP program can be transformed to a UHAT, whose size is still polynomial in $n$. The source of the misunderstanding is probably that we assume $n$ is given in unary as part of a $2^n$-tiling problem instance. Then all representations mentioned above (Turing machine, tiling problem instance, B-RASP, UHAT) have size polynomial in $n$.
> >
> > **Comment/Question:**
> > > This last part -- the proof of Theorem 14 -- is in the best case confusing and in the worst case incorrect ...
> >
> > **Answer:** We agree that having to dig into the proofs of Propositions 6 and 8 to find the constructions of B-RASP and UHAT may lead to confusion. In the revised version of the paper we have added lemmas (Lemmas 8 and 9) that state the construction claims explicitly. The proofs of these lemmas, however, look very similar to the proofs of Propositions 6 and 8 in the original version, since the constructions are the main part there.
> >
> > **Comment/Question:**
> > > Clarification: in section 2.5 (size measures), why is succinctness not defined directly as ...
> >
> > **Answer:** We use a common definition of succinctness, which can e.g. be found as Definition 2.1 in [Grohe & Schweikhardt, 2004], where below the definition also the phrase “exponentially more succinct” is defined. Going through every $f \in 2^{o(n)}$ ensures that there is no subexponential function such that every $R_1 \in C_1$ can be represented by some $R_2 \in C_2$ where the size of $R_2$ is bounded by this subexponential function.
> > In your suggestion what does $R_1(n)$ and $R_2(n)$ mean? We define $R_1$ and $R_2$ as descriptions of languages (e.g. UHATs, LTL formulas, …) and the size of those descriptions is plugged into the function $f$ above as $n$.
> >
> > **Comment/Question:**
> > > The paragraph at the bottom of Example 4 ("This allows us to succinctly ...") is unclear ...
> >
> > **Answer:** Example 4 is meant to illustrate that some B-RASP programs only accept very long strings. In the example we use a string that corresponds (in a simplified form) to one row of the encodings of the $2^n$-tilings, that we want the B-RASP in Proposition 6 to accept. “Stacking multiple such strings vertically” tries to illustrate that the strings from the example can be extended to even longer strings, which corresponds to a tiling with multiple rows. In Theorem 14 we then observe that there are $2^n$-tiling problem instances whose smallest solution has $2^{2^n}$ many rows. How encodings of tilings can be accepted in B-RASP is in detail shown in the proof of Proposition 6 and Example 4 only gives some intuition of the techniques that we use. We have added a clarification of this paragraph to the revised version of the paper.

---

### Author Response · Authors · 2025-11-28

Dear AC and Reviewers,

We thank you for your valuable comments. Due to the unexpected circumstances with OpenReview, we thought a summary of the latest state of the rebuttal would be useful. We would like to emphasize: *We have not attempted (and will never attempt) to discover the identity of our reviewers or communicate outside OpenReview.*

Our initial scores **8, 8, 8, 4** have remained after the rebuttal. Overall, we believe that these reflect our satisfactory response to the reviewers' comments.

More details: one reviewer has responded that we have satisfactorily addressed his/her concerns/questions and kept the score 8. Reviewer ajWw (score: 4) expressed his/her positive comments regarding the contributions, but questioned the technical part of the paper, saying that he/she will raise the score if the questions are satisfactorily addressed. We have explained in our response that this was simply a misunderstanding of some steps in our proofs, and provided thorough answers to the reviewers' questions. Further, we have also uploaded a new version with extra explanations. We would love to hear the reviewer's response, although we understand that this is unfortunately not possible under the current circumstances.

We are extremely grateful for your feedback and the rebuttal process, which has strengthened the paper. We will be happy to answer any additional questions.

Best wishes,

The Authors

---

### Meta-Review · Program_Chairs · 2026-01-09

**Summary:**

The paper proposes a new measure of expressive power for transformers. Below are the reviewer concerns:
aJWw:
1) Some key aspects and assumptions in the proof are missing
2) Some statements should be described more carefully
3) The proof of Theorem 14 (arguably the main result) is confusing

cqAR:
It is shown that UHat are more succint that RNN; but RNN are more expressive, means classes are incomparable -> Title misleading

rC5n:
1) UHAT is a minimal transformer abstraction, how it will transfer to practical architectures - not clear
2) Proposition 6 and 8 omits the UHAT layer-level construction

TYFS:
It is unclear if transformers can learn the structures from Example 4.

**Reviewer Concerns:**

I think all reviewer concerns have been addressed.

**Reviewer Scores:**

The paper has high scores, I would guess it would be 8->8, 8->8, 8->8 and 4->8

---

### Decision · Program_Chairs · 2026-01-26

**Decision:**

Accept (Oral)

**Comment:**

This decision has been updated in view of AC clarifying their decision.